# Extracellular nicotinate phosphoribosyltransferase binds Toll like receptor 4 and mediates inflammation

Antonella Managò[1,12], Valentina Audrito[1,12], Francesca Mazzola[2], Leonardo Sorci [3], Federica Gaudino[1], Katiuscia Gizzi[4], Nicoletta Vitale[5], Danny Incarnato[6], Gabriele Minazzato[7], Alice Ianniello[8], Antonio Varriale[9], Sabato D'Auria[9], Giulio Mengozzi[8], Gianfranco Politano [10], Salvatore Oliviero [4,11], Nadia Raffaelli[7,13] & Silvia Deaglio [1,13]

Damage-associated molecular patterns (DAMPs) are molecules that can be actively or passively released by injured tissues and that activate the immune system. Here we show that nicotinate phosphoribosyltransferase (NAPRT), detected by antibody-mediated assays and mass spectrometry, is an extracellular ligand for Toll-like receptor 4 (TLR4) and a critical mediator of inflammation, acting as a DAMP. Exposure of human and mouse macrophages to NAPRT activates the inflammasome and NF-κB for secretion of inflammatory cytokines. Furthermore, NAPRT enhances monocyte differentiation into macrophages by inducing macrophage colony-stimulating factor. These NAPRT-induced effects are independent of NAD-biosynthetic activity, but rely on NAPRT binding to TLR4. In line with our finding that NAPRT mediates endotoxin tolerance in vitro and in vivo, sera from patients with sepsis contain the highest levels of NAPRT, compared to patients with other chronic inflammatory conditions. Together, these data identify NAPRT as a endogenous ligand for TLR4 and a mediator of inflammation.

[1] Department of Medical Sciences, University of Turin, Turin, Italy. [2] Department of Clinical Sciences, Polytechnic University of Marche, Ancona, Italy. [3] Department of Materials, Environmental Sciences and Urban Planning, Division of Bioinformatics and Biochemistry, Polytechnic University of Marche, Ancona, Italy. [4] Italian Institute for Genomic Medicine, Turin, Italy. [5] Department of Molecular Biotechnology and Health Sciences, University of Turin, Turin, Italy. [6] Department of Molecular Genetics, Groningen Biomolecular Sciences and Biotechnology Institute (GBB), University of Groningen, Groningen, The Netherlands. [7] Department of Agricultural, Food and Environmental Sciences, Polytechnic University of Marche, Ancona, Italy. [8] Department of Laboratory Medicine, Azienda Ospedaliero-Universitaria Città della Salute e della Scienza, Turin, Italy. [9] Institute of Food Science, CNR, Avellino, Italy. [10] Department of Control and Computer Engineering, Polytechnic University of Turin, Turin, Italy. [11] Department of Life Sciences and Systems Biology, University of Turin, Turin, Italy. [12]These authors contributed equally: Antonella Managò, Valentina Audrito. [13]These authors jointly supervised: Nadia Raffaelli, Silvia Deaglio. Correspondence and requests for materials should be addressed to S.D. (email: silvia.deaglio@unito.it)

Pathogen-induced inflammation is triggered by the binding of molecules of bacterial origin to pattern recognition receptors, including Toll-like receptors (TLR)[1,2]. Intrinsic factors produced by the host can modulate this complex network of extracellular signals, thereby contributing to inflammation[3]. Many of these factors have a well-characterized intracellular function and were serendipitously identified in the extracellular space, where they bind and activate pattern recognition receptors[4,5]. Among them are chromatin-associated protein high-mobility group box 1 (HMGB-1), a nuclear DNA binding protein that can be present in the extracellular space through dedicated secretion mechanisms[6,7] and tryptophanyl tRNA synthetase, an intracellular enzyme with a catalytic role in protein synthesis that is rapidly secreted upon pathogen infection and contributes to bacterial clearing[8,9]. The enzyme nicotinamide phosphoribosyltransferase (NAMPT), which catalyzes the first and rate-limiting step in the biosynthesis of NAD from nicotinamide[10,11], is also an important extracellular mediator (eNAMPT). Originally identified as pre-B cell colony enhancing factor (PBEF)[12,13], eNAMPT was later recognized as an essential factor in granulocyte-colony-stimulating factor-(G-CSF)-induced myeloid differentiation[14]. More recently, elevated eNAMPT levels were described in patients characterized by conditions of acute (respiratory distress syndrome) or chronic (type 2 diabetes, obesity, and cancer) inflammation[11,15–17]. eNAMPT effects are mostly linked to the activation of inflammatory programs in macrophages, with recent data suggesting that eNAMPT binds TLR4, adding the enzyme to the number of "danger" signals activating this receptor[16].

NAMPT is structurally and functionally related to the enzyme nicotinate phosphoribosyltransferase (NAPRT), which is rate-limiting in the NAD salvage pathway that starts from nicotinic acid[18,19]. The NAD biosynthetic pathways controlled by NAMPT and NAPRT are closely intertwined and can compensate for each other, as demonstrated by the lack of toxicity of NAMPT inhibitors in cells that express NAPRT[20–22]. NAPRT is localized primarily in the mitochondria and in the cytosol[23], and is believed to boost NAD levels under conditions of cellular stress[24,25].

Starting from the structural and functional similarity between human NAMPT and NAPRT[26], here we investigate whether NAPRT exists in an extracellular form (eNAPRT), thus sharing with NAMPT functional properties that change according to the environment. Our results indicate that NAPRT binds to TLR4, activating the inflammasome and driving transcription of inflammatory cytokines. In addition, eNAPRT regulates monocyte differentiation into macrophages. Sera from patients with sepsis and septic shock contain high levels of NAPRT, underling its potential use as a marker for this critical condition.

## Results

### NAPRT is present in extracellular fluids.
By setting up a luminex assay, we dosed NAPRT in a trial cohort of 25 plasma from normal blood donors (HD), with median concentrations similar to those recorded for eNAMPT (Fig. 1a)[27–29]. Antibody binding to recombinant NAPRT (rNAPRT) and lack of cross-reaction between rNAPRT and recombinant NAMPT (rNAMPT) confirmed the specificity of the assay (Supplementary Fig. 1a, b and Supplementary Table 1). eNAPRT levels were then determined in a validation cohort of 96 HD, including children (age range 2–74 years), indicating median eNAPRT levels of 1.3 ± 0.08 ng/ml (Fig. 1b), with no differences according to gender or age (Supplementary Fig. 1c, d).

To confirm that the protein detected by luminex is NAPRT, two plasma samples containing high levels of the enzyme were immunoprecipitated using an anti-NAPRT monoclonal antibody covalently bound to immunomagnetic beads, revealing a band with a molecular weight of ≈58 kDa, compatible with the NAPRT monomer (Fig. 1c). Next, we used mass spectrometry to identify NAPRT proteotypic peptides in the samples. To this aim, we first created a local spectral library of human NAPRT by trypsin digestion of the recombinant protein and then used a trial serum where rNAPRT was exogenously added at the concentration of 50 ng/ml. Once this approach identified NAPRT peptides, we used a pool of sera containing high levels of eNAPRT, detecting three proteotypic NAPRT peptides (Fig. 1d).

Next, by using a fluorometric assay[30] to measure eNAPRT activity in plasma of 8 HDs, we confirmed that, in the presence of substrates and cofactors, the endogenous enzyme synthesizes nicotinate mononucleotide (NaMN, Fig. 1e). In line with previous data, eNAPRT activity was markedly higher than eNAMPT activity[30].

We then dosed eNAPRT in sera from patients with conditions of acute or chronic inflammation. Plasma was collected from patients with sepsis or septic shock due to bacterial infections (n = 100) who were admitted to the emergency room between October 2016 and April 2017, and were followed-up until January 2018. Sera from 312 patients with a diagnosis of cancer, including solid tumors (lung, prostate and bladder cancer, metastatic melanoma and mesothelioma) and hematological malignancies [chronic lymphocytic leukemia (CLL), myeloma and diffuse large cell lymphoma (DLCL)] were also analyzed by luminex assay. Results indicate that median eNAPRT levels rise from 1.4 ± 0.07 ng/ml in HD to 2.4 ± 0.15 ng/ml in cancer patients to 27.1 ± 4.9 ng/ml in septic individuals (p < 0.0001 for both the comparisons, Fig. 1f), underlying high levels of this enzyme in the latter condition. In contrast, median eNAMPT levels increased from 1.4 ± 0.2 ng/ml of HD to 4.6 ± 0.5 ng/ml in cancer (n = 230, p < 0.0001) to 5.05 ± 4.6 ng/ml in septic patients (n = 100; p < 0.0001, Fig. 1g). Of note, the luminex assay did not recognize bacterial NAPRT (PncB), excluding contamination of PncB in samples derived from septic patients (Supplementary Fig. 1e).

### Extracellular NAPRT drives inflammatory responses.
After observing elevated NAPRT levels in septic patients, we investigated the effects induced by recombinant NAPRT (rNAPRT) in macrophages, which are the mastermind of inflammation. By using RNA sequencing (RNA-seq), we found that rNAPRT exposure for 6 h modulated a total of 1026 genes while recombinant NAMPT (rNAMPT), used for comparison, modulated 626 genes. The majority of genes modulated by rNAMPT (555/626, 88.7%) were also modulated by rNAPRT, while the rNAPRT gene signature was more complex, with only 54.1% of modulated genes shared with rNAMPT (555/1026; Fig. 2a and Supplementary data 1–3). Analysis of the genetic pathways indicated that inflammation, signaling, and immune response were the most commonly enriched biological functions/pathways upon rNAPRT/rNAMPT exposure (Fig. 2b). Heat map in Fig. 2c showed the most upregulated genes upon rNAPRT/rNAMPT treatment belonging to the NF-κB pathway.

In agreement with RNA-seq data, treatment of both human (Fig. 2d and Supplementary Fig. 2a) and mouse (Fig. 2e) macrophages with rNAPRT activated the NF-κB pathway, as determined by western blot analysis showing phosphorylation of the IKKα/β protein, of the p65 subunit and of ERK1/2. rNAPRT effects were dose-dependent, decreasing steadily from 1 μg/ml to 31 ng/ml (Supplementary Fig. 2b). Consistently, confocal microscopy highlighted accumulation of p65 in the nucleus starting 30 min after rNAPRT treatment, peaking at 1 h (Fig. 2f) and decreasing after 6 h (Supplementary Fig. 2c). rNAMPT and LPS

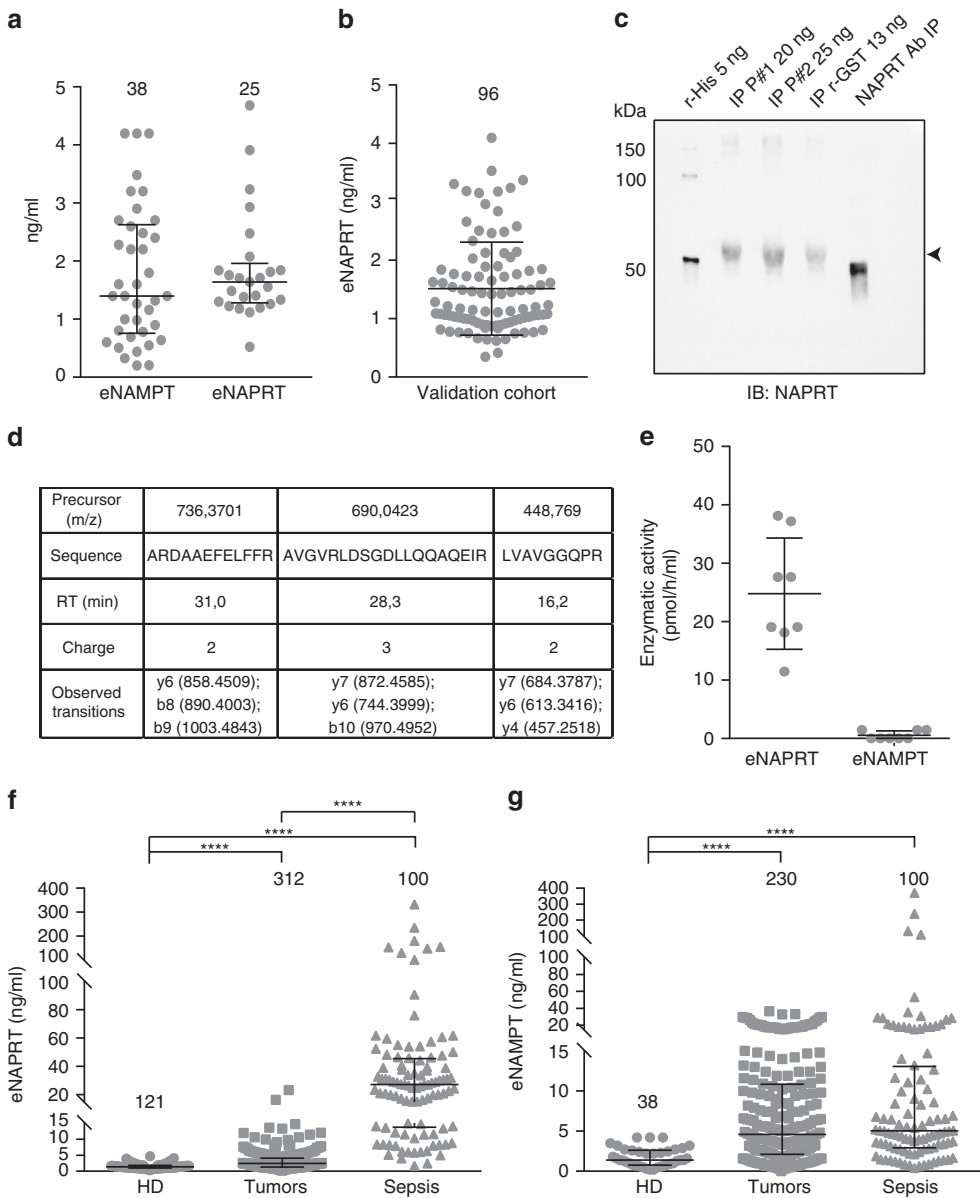

**Fig. 1** eNAPRT is present in normal human plasma, increasing in septic patients. **a** eNAMPT and eNAPRT concentrations (ng/ml) as measured in plasma from HD. **b** eNAPRT concentrations (ng/ml) measured by luminex in a validation cohort of plasma from HD ($n = 96$). **c** The presence of eNAPRT was confirmed by western blot performed on immunoprecipitated (IP) fractions of recombinant rNAPRT-GST-tag (MyBioSource MBS969577) and IP from NAPRT-enriched plasma of two donors (#1–2). rNAPRT-His-tag (home-made recombinant full-length protein) was loaded as control. The protein amount refers to eNAPRT levels measured by luminex. The primary anti-NAPRT antibody used for immunoprecipitation, loaded in the last right lane and recognized by secondary antibody, was an internal control to confirm that samples were not contaminated by primary antibody. **d** Table reporting the three proteotypic NAPRT peptides at the expected retention times (RT), identified by the observed transitions with an error within 15 ppm using a targeted proteomic approach on a pool of sera containing high levels of NAPRT. **e** Graph showing eNAMPT and eNAPRT enzymatic activities (pmol product/h/ml) in the plasma of HD samples, as determined by a fluorometric assay. **f, g** Scatter dot plots showing eNAPRT and eNAMPT levels measured by luminex or quantitative ELISA performed on plasma/sera samples from HD (circles, $n = 121$ for NAPRT and $n = 38$ for NAMPT), tumor patients (squares, $n = 312$ for NAPRT and $n = 230$ for NAMPT) or septic patients (triangles, $n = 100$ for both). Mann–Whitney test. The line in the dot plot defines the median and the error bars define the interquartile range. Source data are provided as a Source Data file

were included as positive controls and both robustly activated NF-κB signaling (Supplementary Fig. 2d). The finding of IRAK1 degradation in response to rNAPRT (Fig. 2g), suggested that this pathway is MyD88-dependent, as also confirmed by significant reduction of NF-κB activation in response to rNAPRT in MyD88-silenced macrophages (Fig. 2h and Supplementary Fig. 2e, f). Induction of the NF-κB-regulated genetic program was confirmed after observing transcription and secretion of pro-inflammatory cytokines, including *IL1B*, *IL8*,

*TNFA*, *CCL3*, and inflammatory mediators such as *CASP1* and *P2RXR* (Fig. 2i and Supplementary Fig. 3a, b). Stabilization of the inflammasome after 6 h of treatment with rNAPRT was documented by increased expression of NLRP3 and Caspase-1 (Fig. 2j and Supplementary Fig. 3c). All data were confirmed in at least five different preparations of macrophages from normal donors.

LPS contamination of our rNAPRT preparations was ruled out on the basis of the following data. First, all proteins were

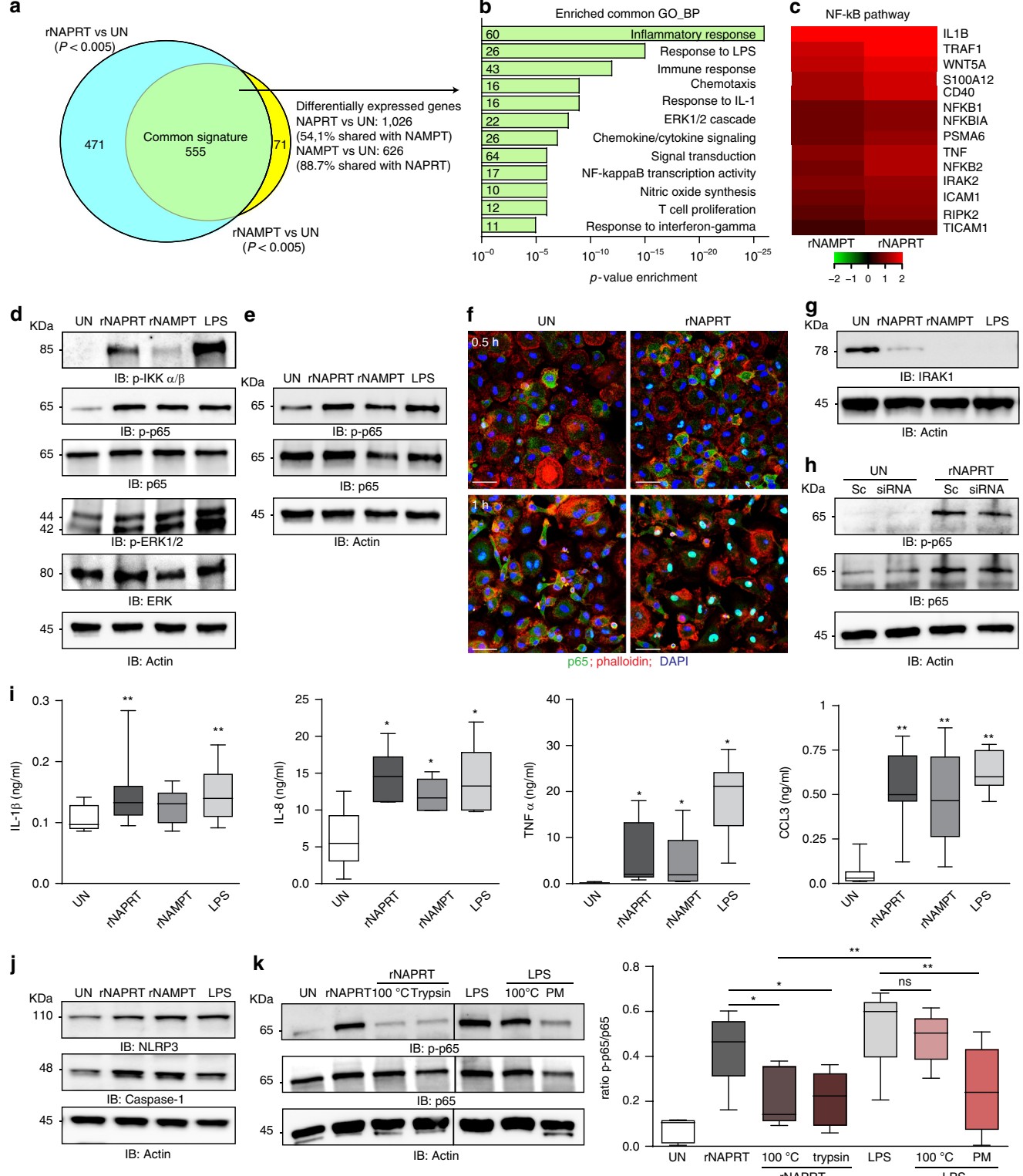

produced in ClearColi, a genetically modified *E. coli* strain that does not trigger endotoxin responses. Second, boiling of rNAPRT for 10 min or protein digestion with trypsin abrogated NF-κB activation, while leaving unaltered LPS-mediated signals. Third, pre-incubation with polymyxin B, an antibiotic that blocks the activity of LPS through binding to lipid A, abrogated signal mediated by LPS (Fig. 2k).

Overall, these data demonstrate that eNAPRT triggers an inflammatory response in macrophages.

**eNAPRT forces monocyte differentiation into macrophages.** rNAPRT and rNAMPT modulated genes encoding for cytokines and chemokines involved in myeloid-macrophage differentiation, including the colony-stimulating factors (*CSF1,2,3*). These cytokines, in particular *CSF1*/M-CSF, induce differentiation of hematopoietic stem cells and circulating monocytes into macrophages, controlling their polarization, phagocytosis, and chemotaxis[31,32]. We confirmed significantly increased M-CSF transcription and secretion upon treatment of normal human

**Fig. 2** eNAPRT induces an inflammatory gene signature in macrophages. **a** Venn diagram showing RNA-seq analysis results in human macrophages treated with rNAPRT or rNAMPT (1 μg/ml, 6 h) and compared to untreated (UN) condition (n = 3). **b** Histograms represent the most significantly enriched gene categories (gene ontology GO, biological processes). The number of genes belonging to each GO is indicated near the y-axis. **c** Heatmaps of the most up-regulated genes by rNAPRT and rNAMPT vs untreated condition, belonging to the "NF-κB pathway" cluster. **d** Western blot analysis of p-IKKα/β, p-p65, pERK1/2 in HD macrophages upon treatment (30 min) with rNAPRT (1 μg/ml), rNAMPT (1 μg/ml), and LPS (2 μg/ml). **e** Western blot analysis of p-p65 in macrophages derived from C57BL/6 wt mice treated as in **d**. **f** Confocal microscopy analysis of p65 staining in human macrophages treated with rNAPRT (1 μg/ml, 0.5 or 1 h) [original magnification was ×63, scale bar: 25 μm, samples from four different HD, at least three different field/slides were counted]. **g** Western blot analysis of IRAK1 in HD macrophages (n = 6) upon treatment (30 min) with rNAPRT (1 μg/ml), rNAMPT (1 μg/ml), or LPS (2 μg/ml). **h** Western blot analysis of p-p65 in scramble (sc) or MyD88 siRNA-silenced macrophages upon treatment as in **d**. **i** Box plots showing protein concentration of IL-1β, IL8, TNFα, and CCL3 evaluated by ELISA in supernatants derived from macrophages (at least n = 6) treated (15 h) with rNAPRT (1 μg/ml), rNAMPT (1 μg/ml), and LPS (2 μg/ml). Wilcoxon test. **j** Western blot analysis of NLRP3 and Caspase-1 in HD macrophages upon treatment (6 h) with rNAPRT (1 μg/ml), rNAMPT (1 μg/ml), and LPS (2 μg/ml, n = 7 for all conditions). **k** Western blot analysis of p-p65 in HD macrophages (n = 5) upon treatment (30 min) with rNAPRT (1 μg/ml) or LPS (2 μg/ml) in different conditions (PM: polymyxin B). Paired t-test. In the box plots the line in the box defines the median and the error bars define the minimum and maximum of all data. Source data are provided as a Source Data file

monocytes with both rNAPRT and rNAMPT (Fig. 3a). Consistently, treatment of PBMC preparations from HDs with rNAPRT resulted in marked increased numbers of adherent cells, with morphologic features of macrophages, as shown by Giemsa staining of residual cells after 10–12 days of culture (Fig. 3b, c). rNAMPT was used as positive control in myeloid-component priming (Fig. 3a–c)[14,27,33]. As expected, LPS did not induce long-term macrophage differentiation, consistent with previous data indicating that it opposes M-CSF functions (Fig. 3a–c)[34,35]. In line with the induction of differentiation following rNAPRT and rNAMPT treatments, macrophages also upregulated lineage-specific markers, including CD11b and CD68 (Fig. 3d). Notably, eNAPRT could be detected in macrophage culture supernatants, suggesting that macrophages are a source of eNAPRT in vivo (Fig. 3e–g).

Overall, these data demonstrate that eNAPRT induces inflammatory responses in macrophages and enhances their differentiation from circulating monocytes.

**eNAPRT binds to TLR4.** To investigate the mechanisms of action of eNAPRT, we first used the NAPRT mutant G379A, which is devoid of catalytic activity[36]. Through gel filtration chromatography, we showed that the mutant co-eluted with the wild-type protein, indicating correct protein folding (Supplementary Fig. 4a). Treatment of macrophages with this mutant triggered p65 phosphorylation and nuclear translocation (Fig. 4a, b and Supplementary Fig. 4b, c). Under these conditions, no differences between the G379A mutant and the wild-type form could be observed, indicating that the enzymatic activity is dispensable for signaling and suggesting that the maintenance of a proper folding might be essential for such a function.

Based on the evidence that (i) eNAMPT binds TLR4[16] and (ii) NAMPT and NAPRT share a high degree of structural similarity[26], we hypothesized that the signaling function of eNAPRT could be also mediated by TLR4. To confirm this, we analyzed the interaction between rNAPRT and the recombinant extracellular domain of TLR4 (rTLR4) through surface plasmon resonance (SPR) under the conditions previously established for the rNAMPT-rTLR4 interaction[16]. By using a surface coated with an anti-NAPRT antibody, we showed that a pre-mixed solution of rNAPRT and rTLR4 resulted in increased binding when compared to rNAPRT alone, indicating that a direct molecular interaction is occurring between the proteins (Fig. 4c and Supplementary Fig. 4d).

To validate these results in a cellular setting, we transiently silenced TLR4 expression in normal human macrophages (Supplementary Fig. 5a, b). Treatment of TLR4-silenced macrophages with rNAPRT resulted in significant decrease of NF-κB activation, as shown by western blot and staining for nuclear p65

(Fig. 4d, e and Supplementary Fig. 5c). Consistently, rNAPRT-driven *IL1B* and *IL8* transcription was severely impaired in TLR4-silenced macrophages, confirming a signaling block (Fig. 4f). As expected, activation following rNAMPT and LPS treatment drastically decreased in TLR4-silenced macrophages (Fig. 4d, e and Supplementary Fig. 5c). To provide a formal validation of TLR4-dependent eNAPRT signaling, we obtained macrophages from TLR4$^{-/-}$ mice and treated them with rNAPRT, rNAMPT, and LPS, observing complete loss of NF-κB activation and significant impairment in transcription of pro-inflammatory cytokines, including *CCL2* and *IL1B* (Fig. 4g, h and Supplementary Fig. 5d).

**Structural determinants of eNAPRT involved in TLR4 binding.** Given the ability of human NAPRT and NAMPT to prime innate immune responses, we asked whether the bacterial orthologs might be endowed with the same capability. Bacterial rNAPRT (PncB) from *Streptococcus pyogenes* or bacterial rNAMPT (NadV) from *Acinetobacter bayly* invariably failed to activate NF-κB signaling in all the macrophage preparations tested (Fig. 5a–c), indicating that the signaling function of human NAMPT and NAPRT is not an evolutionarily conserved trait. Exploiting this finding, we sought to map potential molecular determinants with signaling function by comparing the three-dimensional structures of human and bacterial NAPRT, which are dimers. A structural superposition of the human NAPRT dimer (PDB ID: 4YUB) and the ortholog from *E. faecalis* (PDB ID: 4MZY), chosen as a proxy for the *S. pyogenes* protein used in this study, is shown in Fig. 5d. The proteins share 31% of sequence identity (Supplementary Fig. 6), with a very similar overall architecture reflected in a head-to-tail arrangement of the monomers[26]. Nonetheless, a few interesting differences are evident. The human enzyme has a unique insertion of 46 amino acids which is structurally organized in a loop-helix-loop and is exposed to the solvent (Fig. 5d and Supplementary Fig. 6). This region accounts for the length difference between the two proteins, being human NAPRT of 538 amino acids and the bacterial ortholog of 496 amino acids, respectively. Furthermore, a comparison of the surface properties of the two proteins revealed the presence in human NAPRT of an arginine-rich stretch ($_{65}$**R**FL**R**AF**R**L**R**) forming a large mouth-like positively charged area on the top of the dimer (Fig. 5e), which is absent in the bacterial ortholog. None of such arginine residues are involved in the dimer stabilization, and most of the lateral chains are exposed to the solvent (Supplementary Fig. 7a). Notably, the corresponding region in the NAMPT dimer is also positively charged, with arginine replaced by lysine (Supplementary Fig. 7a). For a preliminary validation of the in silico predictions, we generated a NAPRT mutant characterized by replacement of arginines at positions 65, 68, 71, and 73 with

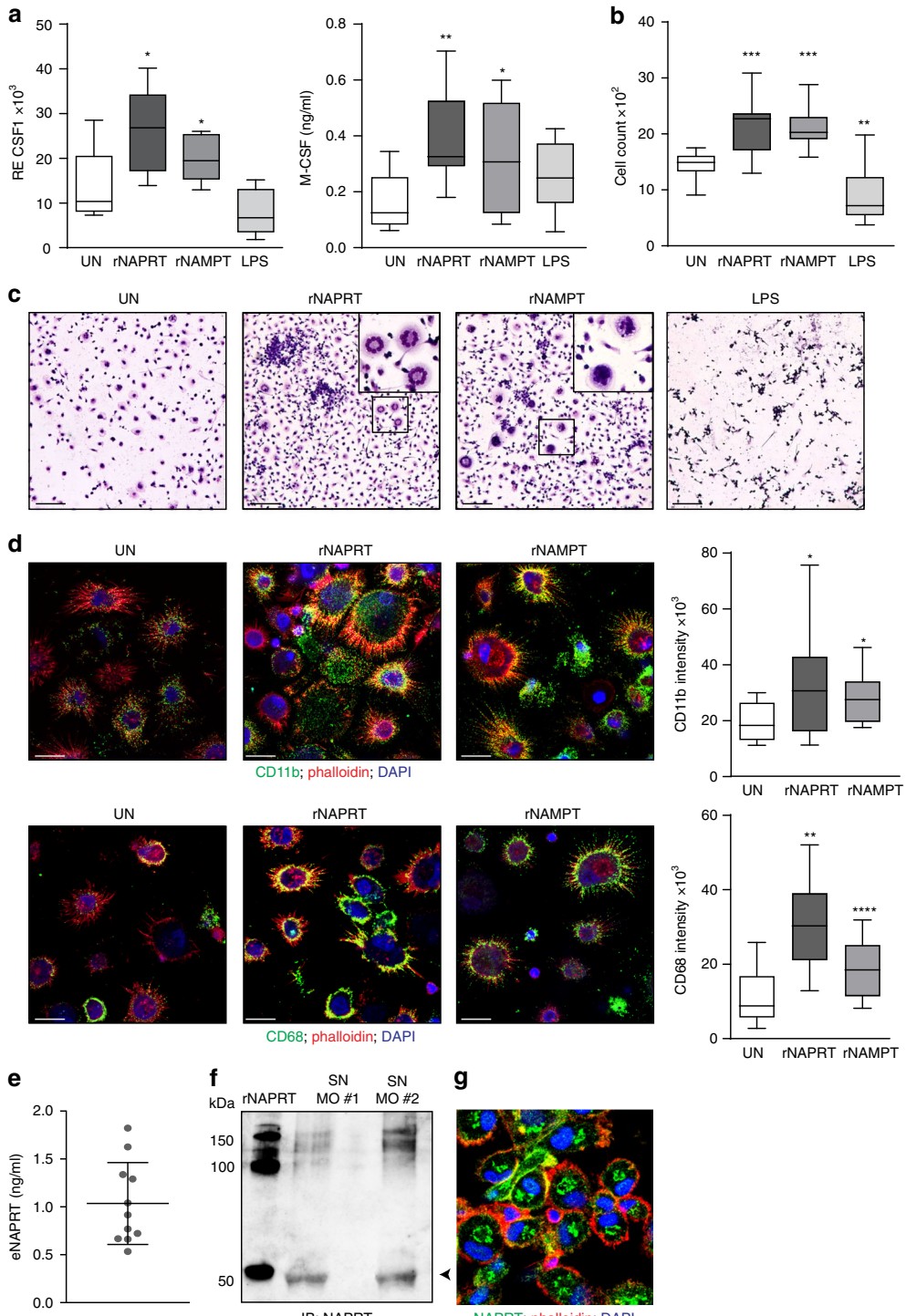

alanines (Supplementary Fig. 7b). This mutant failed to activate NF-κB and to induce cytokines production (Fig. 5f–g), indicating that the arginine-rich region is involved in the binding to TLR4. The correct folding of the mutated protein was assessed through gel filtration chromatography (Supplementary Fig. 7b).

**eNAPRT is a novel risk factor in patients with sepsis.** We then observed that cellular stress is accompanied by marked increase in NAPRT in the culture media (Fig. 6a). Specifically, treatment of human macrophages with TNF-α and cycloheximide to trigger

apoptosis, or with ionomycin and carbonyl cyanide 3-chlorophenylhydrazone (CCCP) to trigger necrosis, induced rapid and significant release of NAPRT, as previously described also for HMGB-1[6] and others DAMPs[37].

Lastly, to investigate the relationship between NAPRT and LPS, normal Balb/c mice were treated with rNAPRT alone or in combination with LPS. rNAPRT was administered to Balb/C mice at 1 mg/kg (low dose) or 25 mg/kg (high dose), in keeping with doses previously used with other DAMPs[9,38]. No signs of toxicity or lethality were observed with low doses, while mice treated with high doses rNAPRT developed signs of endotoxemia,

**Fig. 3** eNAPRT enhances differentiation of myeloid cells into macrophages via M-CSF secretion. **a** Box plots showing mRNA expression levels of *CSF1* (left panel) in macrophages (n = 5) treated with rNAPRT (1 µg/ml), rNAMPT (1 µg/ml), and LPS (2 µg/ml, left panels) for 15 h (paired *t*-test) and M-CSF protein concentration in corresponding culture supernatants measured by ELISA (n = 8, right panel, Wilcoxon test). **b** Box plot representing the cell count of residual adherent cells obtained from HD PBMC preparations (n = 8) treated once at the beginning of culture with rNAPRT, rNAMPT, or LPS (all at 10 ng/ml). After 10–12 days of culture, cells were stained with Giemsa. Representative images (×10, scale bar: 200 µm, and a zoomed area in the square) are shown in **c**. Images were acquired using a CANON EOS 600D camera fitted to an AXIO Lab A1 ZEISS microscope. Cell count was calculated with ImageJ software (freely downloadable at http://rsbweb.nih.gov/ij/, paired *t*-test). **d** Confocal microscopy analysis of the expression of CD11b and CD68 in residual adherent cells obtained in the same conditions as in **c** (original magnification ×63, scale bar: 10 µm). Cells were counter-stained with Alexa568-phalloidin and DAPI. On the side, box plots indicating the relative mean fluorescence intensity of three different experiments. Mann–Whitney test. **e** eNAPRT protein concentration measured in macrophage supernatants by NAPRT luminex assay (n = 11) (**f**) and western blot performed on concentrated (×10) culture supernatants derived from two different HD macrophage preparation (SN MO). rNAPRT was loaded in the gel as control. **g** Confocal microscopy analysis of NAPRT protein expression in macrophages (original magnification ×63, the image represents a zoomed area). Cells were counter-stained with Alexa568-phalloidin and DAPI. Results are reported as box plots and dot plot, where the line in the box defines the median and the error bars define the minimum and maximum or the interquartile range. Source data are provided as a Source Data file

including piloerection, lethargy, diarrhea and conjunctivitis, within 2 h since treatment. However, none of these animals died, at variance with what observed when administering high doses of LPS (25 mg/kg): in these conditions all mice died between 20 and 36 h (Fig. 6b). We then combined low doses of rNAPRT with a lethal dose of LPS, and observed significantly reduced signs of endotoxemia. Importantly, all animals survived with this treatment schedule (6/6, p = 0.0007 Log-rank test, Fig. 6b), suggesting that low doses of NAPRT mediate endotoxin tolerance. No survival advantage was observed when combining rNAPRT (25 mg/kg) with a lethal dose of LPS (Fig. 6b). Accordingly, in vitro pre-treatment with low dose rNAPRT significantly decreased the amount of *TNFA* transcription upon high dose LPS exposure (Fig. 6c and Supplementary Fig. 8a), suggesting that rNAPRT mediates endotoxin tolerance[39]. With this information, we re-examined our cohort of septic patients and noticed that patients who died because of septic shock had significantly higher levels of eNAPRT in the plasma (Fig. 6d), while those with low concentrations survived. This finding suggests that "physiological" levels of eNAPRT may be essential to prevent mortality in response to bacterial infections. At variance, no differences in eNAMPT levels were observed (Supplementary Fig. 8b). Next, by exhaustively constructing a set of confusion matrices, we determined the best cut-off value for eNAPRT at 15 ng/ml (p = 0.001 Fisher's exact test). Using this cut-off, we observed that in the eNAPRT ≥ 15 subset 31/71 (44%) patients died because of septic shock, compared to 3/29 (10%) patients in the eNAPRT < 15 ng/ml counterpart (p < 0.0001 Fisher's exact test; Fig. 6e). Furthermore, patients with eNAPRT levels ≥15 ng/ml were characterized by worse renal function (median creatinine levels 1.8 ± 0.14 vs 1.1 ± 0.29, p = 0.01), higher lactate dehydrogenase (median LDH levels 488 ± 48 vs 392 ± 53, p = 0.04) and C-reactive protein (median CRP levels 209 ± 14 vs 130 ± 18, p = 0.006), in line with a compromised clinical picture (Fig. 6f). Risk analysis showed that septic patients with eNAPRT levels ≥ 15 ng/ml had a 4.46-fold increased risk of mortality, compared to patients with eNAPRT levels < 15 ng/ml ([CI 95%: 1.46; 13.70], p = 0.001 $\chi^2$ test, Supplementary Fig. 8c). Kaplan–Meier curves confirmed that the overall survival of patients with eNAPRT levels ≥15 ng/ml was markedly shorter than the counterpart (p = 0.005, Log-rank test, Fig. 6g).

Lastly, we tried to better stratify mortality risk by determining which factors associate to eNARPT, increasing its predictive value. Biochemical parameters commonly measured during routine screening for infectious patients were considered, including CRP, procalcitonin (PCT), white blood cells count (WBC), platelets (Plts), International Normalized Ratio (INR) derived from prothrombin time (PT) and eNAMPT. The most noticeable combined effect was between eNAPRT and

CRP (p = 0.001, Anova test, Supplementary Fig. 8d), also confirmed by a linear regression in 92 plasma samples of septic patients (r = 0.31, p = 0.002, Fig. 6h). An independent validation cohort of 71 septic patients confirmed high levels of eNAPRT in this inflammatory condition, as well as a statistically significant association between high levels of eNAPRT and mortality, confirming that eNAPRT is a new risk factor in sepsis (Supplementary Fig. 8e–g).

## Discussion

Damage-associated molecular patterns (DAMPs) are molecules that can be actively or passively released by injured tissues and that activate the immune system[37]. Here we show that the intracellular NAD biosynthetic enzyme NAPRT is physiologically present at low levels in human sera and that its levels rise sharply in patients with sepsis or septic shock. In keeping with the hypothesis that eNAPRT may act as a new DAMP, we show that it is a potent mediator of inflammatory responses. When added to cultures of human and murine macrophages, rNAPRT rapidly and robustly activates NF-κB and induces a genetic signature of inflammation, with ~1000 modulated genes. Following NAPRT treatment, we documented active synthesis and secretion of pro-inflammatory cytokines, such as IL-1β, IL8, and TNF-α, as well as assembly of the inflammasome.

Because NAPRT is an enzyme involved in the generation of NAD starting from nicotinic acid, which can be present in human plasma, we first asked whether these effects are dependent on the enzymatic activity. To do so, we generated a single amino-acid mutant, which retains the 3D structure of NAPRT, but lacks the enzymatic activity. This mutant is still able to fully activate macrophages, indicating that the enzymatic activity is irrelevant in the pro-inflammatory functions of eNAPRT and pointing to the existence of an eNAPRT receptor. In its quest, we focused on pattern recognition receptors, starting from TLR4, which is one of the main ligands for many DAMPs[37]. Surface plasmon resonance confirmed a direct physical interaction between the extracellular domain of TLR4 and rNAPRT in vitro. In addition, human macrophages, where TLR4 expression had been silenced, were unable to activate NF-κB via rNAPRT, as well as to upregulate expression of NF-κB controlled cytokines, such as IL-1β and IL8. These results indicate that eNAPRT requires TLR4 to signal. Accordingly, macrophages from TLR4$^{-/-}$ mice failed to respond to rNAPRT, both in terms of NF-κB activation and cytokine production.

NAPRT is an evolutionarily conserved protein[40] with the human and the bacterial enzyme sharing a very similar fold, with only minor structural differences[26]. On the basis of our finding that bacterial NAPRT is unable to activate NF-κB, we carried out an in silico study to point out structural determinants possibly

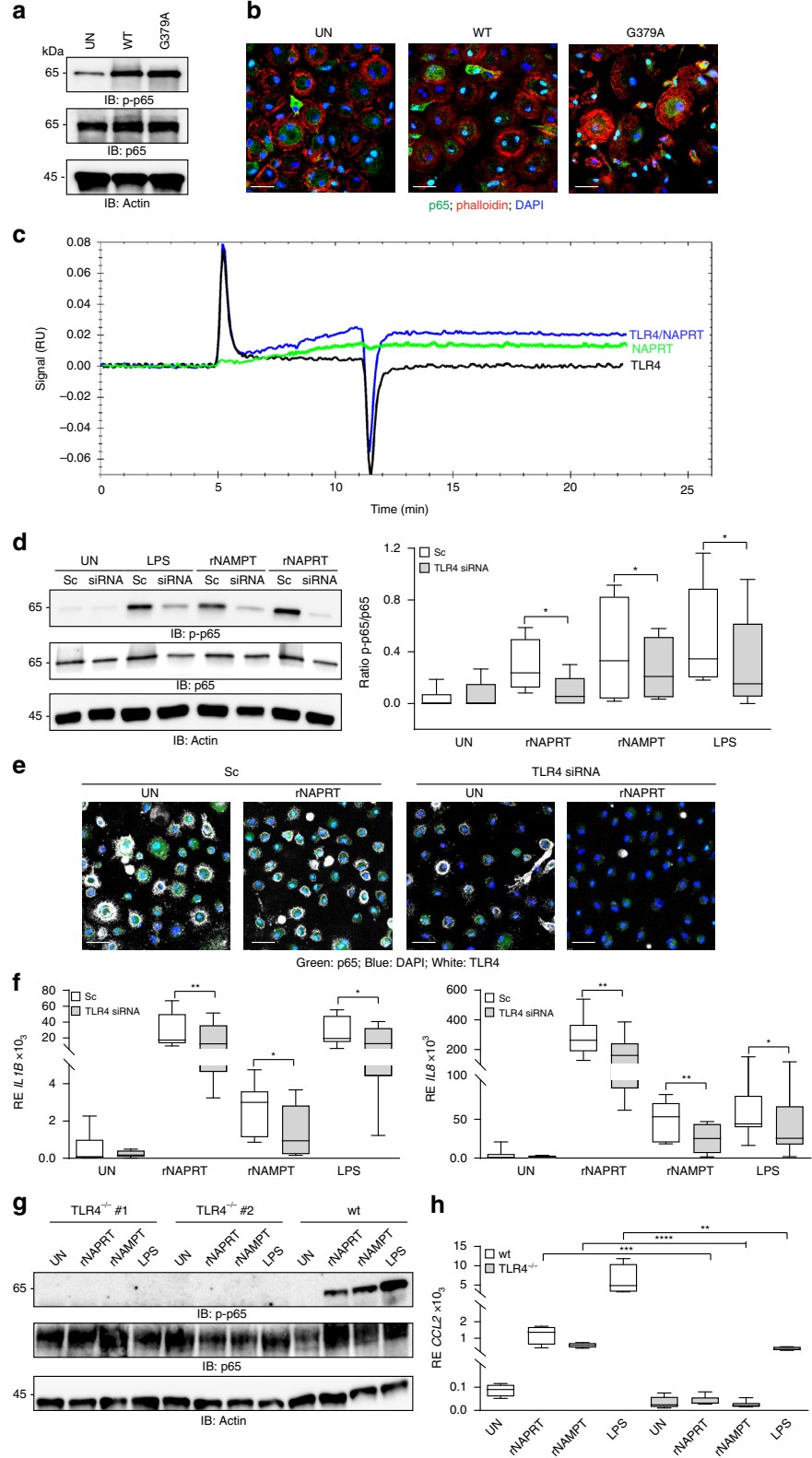

Green: p65; Blue: DAPI; White: TLR4

involved in TLR4 binding. We identified a large arginine-rich region at the surface of the human protein, absent in the bacterial ortholog, as a likely candidate for signaling. A specific mutant, where the arginine residues were replaced with alanine, failed to activate NF-κB signaling, while retaining 3D structure, thus validating our prediction. In keeping with this result, arginine-rich domains of proteins are known to stabilize macromolecular structures through several type of interactions[41], and arginine/lysine residues are often involved in the formation and stability of productive TLR complexes[42].

Among the biological processes commonly regulated by NAPRT and NAMPT are those involved in macrophage recruitment and differentiation, including *CSF1*, which is known to be blocked by LPS[34]. Consistently, treatment of normal human

**Fig. 4** eNAPRT binds to TLR4. **a** Western blot analysis of p-p65 in HDs macrophages upon treatment (30 min) with rNAPRT (WT) or the G379A mutant at the concentration of 1 µg/ml ($n = 5$). **b** Confocal images showing p65 localization (green fluorescence) in human macrophages treated with rNAPRT (WT) or the G379A mutant (1 µg/ml, 30 min). Original magnification ×63, scale bar: 25 µm. **c** SPR measurements of TLR4 (1 µM), NAPRT (100 µM), and TLR4/ NAPRT (1 µM/100 nM). The measurements were performed at 25 °C and the flux was fixed at 30 µL/min. **d** Western blot analysis of p-p65 in scramble (sc) or TLR4 siRNA-silenced macrophages upon treatment (30 min) with rNAPRT (1 µg/ml), rNAMPT (1 µg/ml), and LPS (2 µg/ml). Box plot on the right represents quantification obtained by ImageQuant of p-p65/p65 ($n = 6$ for rNAPRT and LPS, $n = 4$ for rNAMPT, paired $t$-test). **e** Confocal images showing p65 localization (green fluorescence) in macrophages ($n = 3$) transfected with a scramble control siRNA (sc) or a TLR4 siRNA (white fluorescence for TLR4 staining) and treated (30 min) with rNAPRT (1 µg/ml). Original magnification ×63, scale bar: 50 µm. **f** Box plots showing mRNA expression levels of *IL1B* and *IL8* evaluated by qRT-PCR upon 15 h treatments with rNAPRT (1 µg/ml), rNAMPT (1 µg/ml), and LPS (2 µg/ml) of macrophages transfected for 72 h with a scramble (sc) control siRNA or with a TLR4 siRNA ($n = 8$ for rNAPRT and LPS, $n = 6$ for rNAMPT, paired $t$-test). **g** Western blot analysis of p-p65 in macrophages derived from TLR4$^{-/-}$ ($n = 7$) or wt ($n = 5$) mice treated as in **d**. **h** Box plots showing mRNA expression levels of *CCL2* evaluated by qRT-PCR in macrophages derived from TLR4$^{-/-}$ ($n = 6$) or wt ($n = 4$) mice treated with rNAPRT (1 µg/ml), rNAMPT (1 µg/ml), and LPS (2 µg/ml) for 15 h, unpaired $t$-test. Results are reported as box plots, where the line in the box defines the median and the error bars define the minimum and maximum of all data. Source data are provided as a Source Data file

PBMC with rNAPRT significantly enhanced differentiation to full-fledged macrophages, with typical morphological features and expression markers. These cells also expressed and released eNAPRT, suggesting that there may be an autocrine functional loop. This second, long-term function of eNAPRT suggests that it may have a more complex role, not only related to inflammation, but also to tissue repair, as recently described for other DAMPs, including HMGB-1[7].

In vivo, high levels of NAPRT are associated with an unfavorable outcome in septic patients. Our data, obtained in a trial cohort and validated in a second cohort, show that patients with levels of eNAPRT ≥15 ng/ml are characterized by overall worse clinical parameters with 4.46-fold increased risk of mortality, compared to the counterpart and suggest that eNAPRT is a novel risk factor for sepsis. Even if the biological explanation behind this observation is still partly missing, there are significant starting points. First, eNAPRT is markedly increased in the supernatant of cells undergoing necrosis, suggesting that this could be the source in patients with sepsis or septic shock. In fact, eNAPRT levels were significantly higher in the latter condition (median eNAPRT levels were $13.6 \pm 1.7$ ng/ml in patients with sepsis vs $24.7 \pm 2.2$ ng/ml in patients with septic shock, $p = 0.0084$, Supplementary Fig. 8h). Second, low doses of NAPRT prevent LPS signaling in vitro and LPS-induced mortality in mice, in keeping with what observed with other DAMPS, including tRNA synthetase[9]. Importantly, this does not happen when rNAPRT is administered at high doses, suggesting that there are concentration-dependent effects.

Several issues remain to be addressed. First and foremost, it will be important to understand the relationship between eNAPRT and eNAMPT: our findings imply that they may have different roles in acute vs chronic inflammation, engaging TLR4 in different pathological conditions and alerting the immune system to different sets of "dangers". In addition, the mechanisms of release/secretion of the molecule, of its interplay with TLRs and of its role in acute and chronic inflammatory states in vivo deserve further attention. This will be the focus of future investigations.

## Methods

**Patient samples.** Patient plasma/sera were collected in accordance with the Institutional Review Board and the Declaration of Helsinki.

Plasma from healthy donors were obtained from the local Blood Bank.

Sera from all patients with a diagnosis of sepsis and septic shock admitted to the emergency room of the Città della Salute e della Scienza Hospital (Ethical committee of the Città della Salute e della Scienza Hospital, Turin, Italy) were used.

Samples from patients with cancer were provided by clinicians, cited in the "Acknowledgements" section.

**Antibodies used for western blot and immunoprecipitation.** The following antibodies were used for western blot: anti-NAMPT (A300-779A, Bethyl

Laboratories, Montgomery, TX), anti-NAPRT1 (NBP1-87243, Novus Biologicals, Littleton, CO; AMAB90725 Atlas; 66159-I-Ig ProteinTech, Manchester, UK and MBS1491066 MyBioSource), anti-phospho-p65 (Ser536, #3033S), anti-p65 (#8242S), anti-phospho-IKKα/β (Ser176/180, #2697), anti-IRAK1 (#4504), anti-NLRP3 (#13158), anti-Caspase-1 (#3866) and Cyclophilin A (#2175) all from Cell Signaling Technologies (Danvers, MA), anti-pERK1/2 (pT202/Y204, 612358) and anti-panERK (610123) both from BD Biosciences (East Rutheford, NJ), anti-MyD88 (sc-11356, Santa Cruz Biotechnology, Dallas, TX) and anti-actin horse-radish peroxidase (HRP)-conjugated (ab20272, Abcam, Cambridge, UK). Secondary reagents were: goat anti-mouse IgG-HRP-conjugated (Perkin Elmer, Waltham, MA), goat anti-rabbit HRP-conjugated (Santa Cruz Biotechnology).

**Antibodies used for confocal microscopy.** Antibodies used for immunofluorescence were: anti-NAPRT1 (NBP1-87243, 1:100 Novus Biological) anti-p65 (sc-8008, 1:100 from Santa Cruz Biotechnology), anti-TLR4 (NB-10056566, 1:100 from Novus Biotechnology), anti-CD68 AlexaFluor-488 (333812, 5 µl for well from Biolegend, San Diego, CA), anti-CD11b (HPA002274, 1:200 from Sigma-Aldrich, Saint Louis, MO). Secondary reagents were: goat anti-mouse IgG AlexaFluor488-conjugated (1:50) and goat anti-rabbit IgG AlexaFluor488-conjugated (1:100, both from Thermo-Fisher, Waltham, MA). After the primary and secondary antibodies, cells were counter-stained with AlexaFluor 568-conjugated phalloidin (1:100) and 4′,6-Diamidino-2-phenylindole (DAPI, 1:30,000, both from Thermo-Fisher).

**Preparation of recombinant proteins.** Plasmids for human recombinant NAPRT and G379A mutant are described in[36]. Plasmid for the R(65->73)A mutant was obtained by site-directed mutagenesis using QuikChange Lightning kit (Agilent Technologies, Santa Clara, CA). For the expression of *Streptococcus pyogenes* PncB (NAPRT ortholog) the coding region of the *pncB* gene was amplified from genomic DNA and cloned into the pET28a vector at *Nhe*I and *Eco*RI sites. Plasmids for the expression of human NAMPT and its ortholog *Acinetobacter bayly* NadV are described[40,43]. All proteins were expressed in ClearColi BL21(DE3) cells (Lucigen, Middleton, WI) and purified by Ni-NTA affinity chromatography. His Trap columns (GE Healthcare, Chicago, IL) were equilibrated with 100 mM potassium phosphate pH 8.0, 300 mM KCl for NAPRTs and PncB and in 50 mM Hepes pH 7.5, 500 mM NaCl for NAMPT and NadV. After a washing step with 40 mM imidazole in the same buffers, elutions were carried out with a linear gradient up to 350 mM imidazole. PD-10 columns (GE Healthcare) were then used to replace imidazole with 20% glycerol.

**Luminex assay to quantify eNAPRT.** To measure eNAPRT we set-up, in collaboration with Bioclarma, Turin (http://www.bioclarma.com), a new assay exploiting luminex technology and using highly specific monoclonal and polyclonal antibodies. Commercially available rNAPRT-GST-tag (MyBioSource MBS969577) was used to build a titration curve. The detection range of the assay is from 10 pg/ml to 500 ng/ml (Italian patent I0174545 and PCT/IB2019/051314; Inventors: Deaglio S., Audrito V.; Owners: University of Turin & IIGM). Schematic representation of the assay is shown in Supplementary Fig. 9.

**eNAMPT quantification assay.** eNAMPT concentrations in plasma and culture supernatants were determined using human NAMPT Enzyme-Linked Immunosorbent Assay (ELISA) kit (Adipogen, Liestal, CH) and also using Bio-Plex Pro Human Diabetes Assay panel (Bio-Rad, Hercules, CA) that includes NAMPT[29].

**Gel filtration chromatography.** Gel filtration chromatography was carried out to determine correct protein folding, comparing pure rNAPRT to the G379A mutant or R(65->73)A NAPRT mutant. A fast protein liquid chromatography (FPLC, Superose 12 10/300 GL column, GE Healthcare) was used, and the samples were eluted with 100 mM potassium phosphate buffer, pH 8.0, 300 mM NaCl.

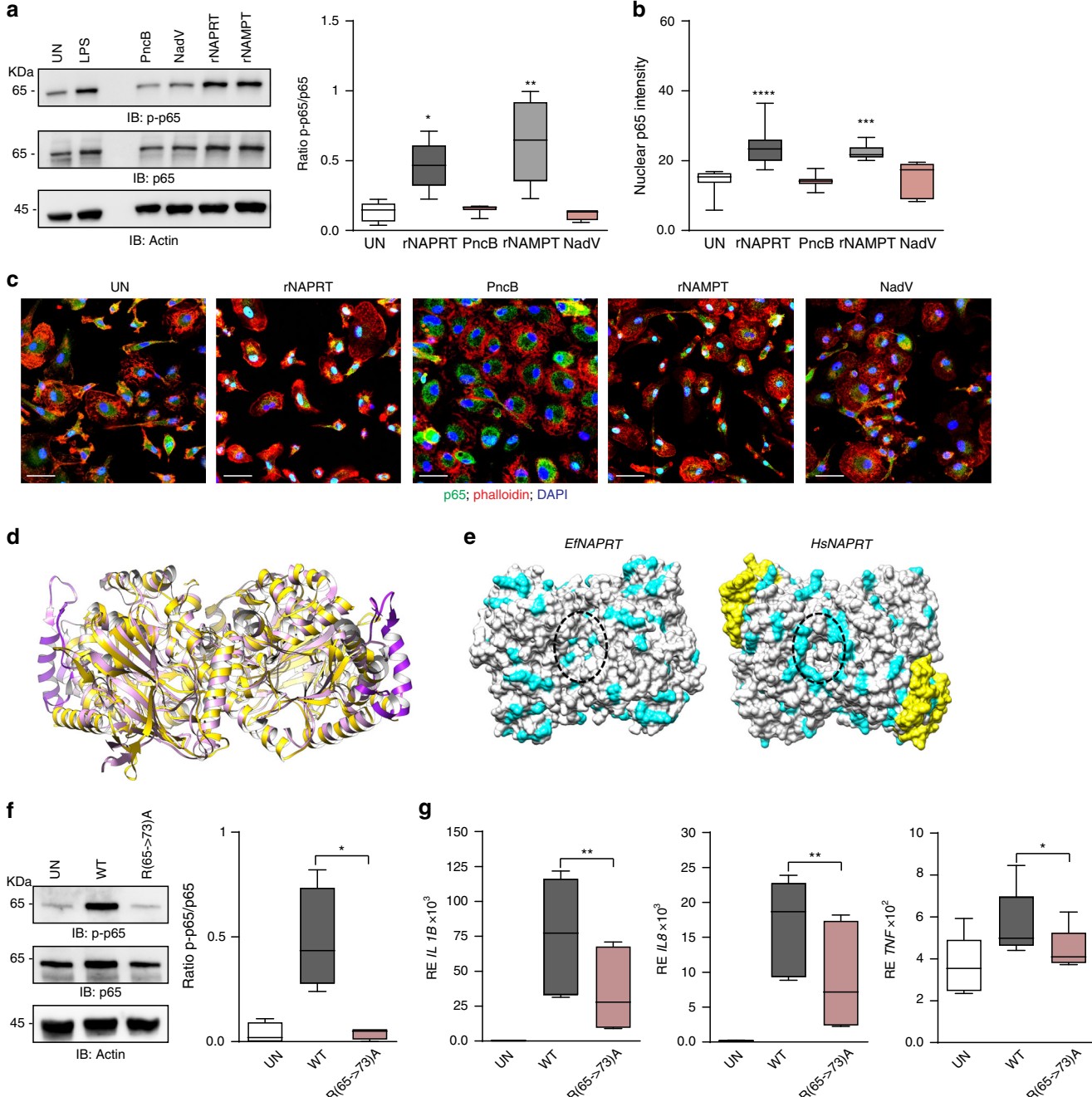

**Fig. 5** Structural determinants of rNAPRT and rNAMPT involved in TLR4 binding. **a** Western blot analysis of p-p65 in HDs macrophages upon treatments (30 min) with rNAPRT, rNAMPT, and their bacteria ortholog, PncB ($n = 7$) and NadV ($n = 4$), respectively (all at 1 μg/ml). Box plot on the right represents band quantification using ImageQuant software of p-p65/p65, Wilcoxon test. **b** Box plot showing relative quantification of nuclear p65 mean fluorescence intensity in human macrophages after 30 min treatments with rNAPRT, PncB, rNAMPT, and NadV (all at 1 μg/ml, at least $n = 3$, Mann–Whitney test). **c** Representative confocal microscopy images showing p65 localization (green fluorescence, original magnification ×63, scale bar: 25 μm). Cells were counter-stained with Alexa568-phalloidin and DAPI. **d** Superposition of dimeric human (Hs) (in pink, PDB: 4YUB) and E. faecalis (Ef, in gold, PDB: 4MZY) NAPRT in ribbon representation. The 46 amino acids insertion of HsNAPRT is highlighted in magenta. **e** The protein surfaces (top view) of HsNAPRT (PDB ID: 4YUB) and EfNAPRT (PDB ID: 4MZY) dimers are colored in white. The positively charged amino acids lysine and arginine are colored in cyan. The 46 amino-acid insertion of HsNAPRT is highlighted in yellow. The mouth-shaped areas of the two enzymes are contoured by a dotted oval. **f** Western blot analysis of p-p65 in HDs macrophages upon treatments (30 min at 1 μg/ml) with wt rNAPRT and R(65->73)A NAPRT mutant ($n = 4$). On the left, box plots represent band quantification using ImageQuant software of p-p65/p65, paired t-test. **g** Box plots showing mRNA expression levels of IL1B, IL8, and TNFA evaluated by qRT-PCR in RNA from HD macrophages ($n = 8$) treated with wt rNAPRT (1 μg/ml) and R(65->73)A NAPRT mutant (1 μg/ml) for 15 h, paired t-test. Results are reported as box plots, where the line in the box defines the median and the error bars define the minimum and maximum of all data. Source data are provided as a Source Data file

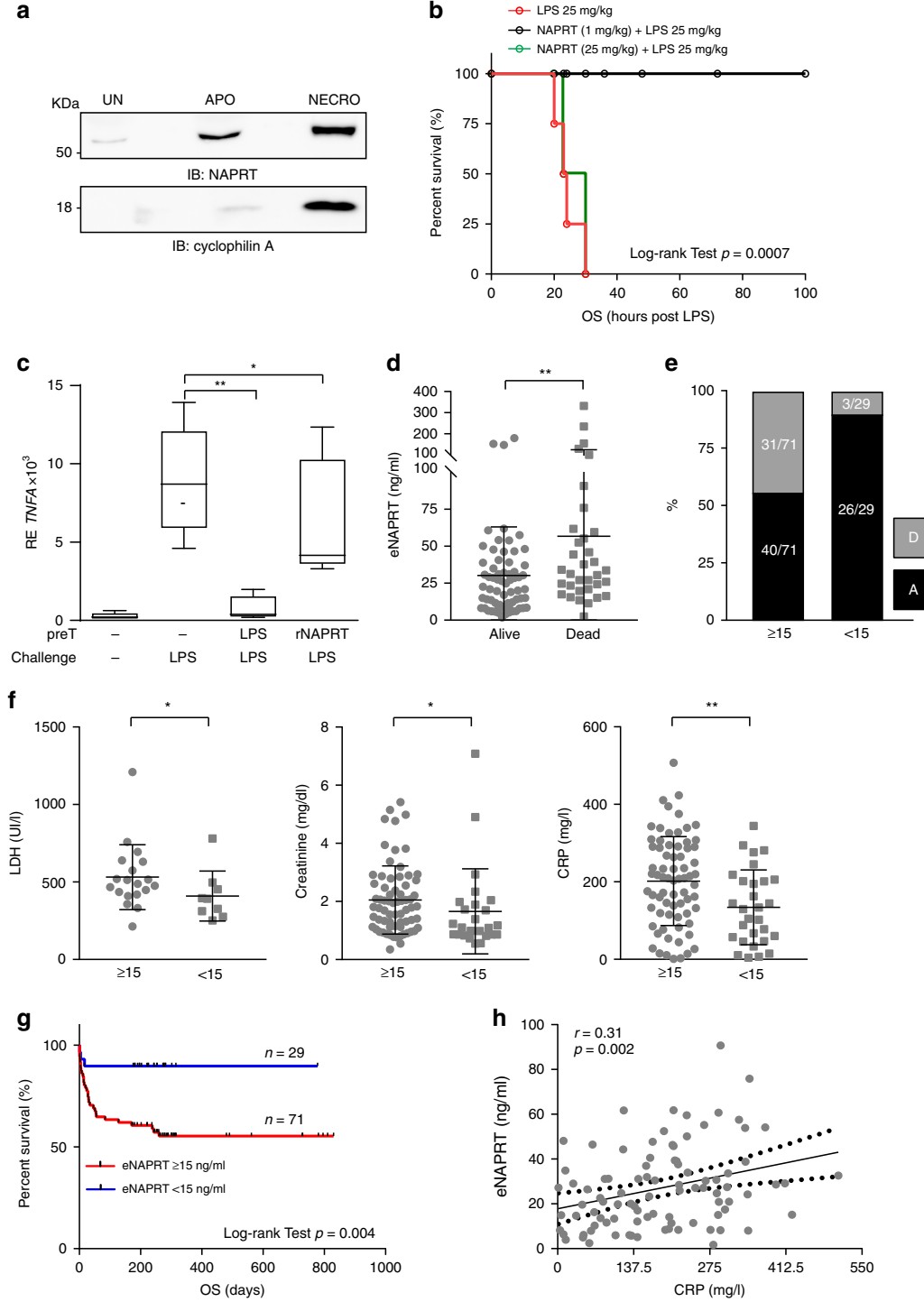

**Fig. 6** eNAPRT is a risk factor for septic patients. **a** Western blot analysis of NAPRT in 10× concentrated HD macrophages supernatants (SN), in which apoptosis (APO) and necrosis (NECRO) were induced as described in "Methods" section. Detection of cyclophilin A in SN was used as positive control for necrosis induction. **b** Kaplan–Meier curves showing overall survival (OS) of mice treated intraperitoneally with LPS (25 mg/kg) with or without rNAPRT (1 mg/kg) or rNAPRT (25 mg/kg) 1–2 h prior to LPS administration. $N = 6$ mice/treatment group. Log-rank test shows statistical significance. **c** Box plots showing mRNA expression levels of *TNFA* evaluated by qRT-PCR in RNA from HD macrophages ($n = 5$) treated as follow: 30 h of pre-conditioning (preT) with LPS 10 ng/ml or rNAPRT 10 ng/ml followed by 6 h of challenging with LPS 1 μg /ml. Paired *t*-test. **d** Scatter dot plot showing eNAPRT levels in septic patients, according to outcome. **e** Graph representing the percentage of septic patients who survived (black bars) or died (gray bars) when dividing patients according to the eNAPRT cut-off of 15 ng/ml. **f** Scatter dot plots reporting LDH, creatinine and CRP levels in septic patients dividing according to the eNAPRT cut-off. Mann–Whitney test. **g** Kaplan–Meier curves showing overall survival (OS) of the cohort of 100 septic patients divided on the basis of eNAPRT levels. Log-rank test shows statistical significance. **h** Regression line showing a positive correlation between CRP (*x*-axis) and eNAPRT (*y*-axis) levels, as detected in 92 plasma from septic patients. Pearson coefficient (*r*) and the corresponding *p*-value are noted. Results are reported as box plots and dot plot, where the line in the box defines the median and the error bars define the minimum and maximum or the interquartile range. Source data are provided as a Source Data file

**Determination of NAMPT and NAPRT activities**. The activity of eNAMPT and eNAPRT was determined by a multi-coupled fluorometric assay[30].

**Proteomics**. Proteomic analyses were performed by the Protein Microsequencing Facility (ProMiFa) of the San Raffaele Scientific Institute, Milan, Italy. The strategy to confirm that the protein identified by luminex was NAPRT was to enrich normal human plasma containing < 1 ng/ml of eNAPRT with human rNAPRT, to reach the final concentration of 50 ng/ml. This plasma was analyzed as is or was depleted of 20 abundant proteins using the ProteoPrep 20 spin column (Sigma-Aldrich). These two samples were prepared with the aim of verifying the feasibility of the identification of NAPRT in a real sample of plasma containing a similar concentration of the protein. The latter was obtained by pooling plasma coming from 15 donors with high level of eNAPRT ($\cong$50 ng/ml).

The immunodepletion procedure for each sample was performed on 8 μl of plasma each time (6 times per sample) and twice repeated in order to get ~99% depletion. The recovered supernatant was analyzed to determine total protein concentration using Direct Detect IR spectrophotometer (Merck-Millipore, Burlington, MA) and bovine serum albumin (BSA) as standard. Total proteins (40 μg) were in-solution digested using Filter Aided Sample Preparation (FASP) protocol as reported in literature[44]. Samples were desalted using C18 home-made tip columns (C18 Empore membrane, 3 M) and injected in a nano UPLC system (Easy nLC-1000, Proxeon Biosystem, Odense, Denmark). Peptide separations occurred on a home-made 12.5 cm reverse phase spraying fused silica capillary column, packed with 1.9 μm ReproSil Pur 120 C18-AQ (DrMaisch GMBH, Ammerbuch, Germany). A gradient of eluents A (pure water with 0.1% v/v formic acid) and B (ACN with 0.1% v/v formic acid) was used to achieve separation (300 nl/min flow rate) from 0 to 45% B in 45 min. Parallel reaction monitoring analyses were performed using a Q-Exactive mass spectrometer (Thermo-Fisher). The acquisition method combined two scan events corresponding to a full scan MS [resolution set at 35,000 mass to charge ($m/z$) 200] and a PRM event (resolution set to 17,500 at $m/z$ 200; isolation window set to 2 $m/z$; maximum fill time of 120 ms and normalized collision energy set to 27) which targeted the precursor ions of the peptides at their relevant charge states in a 1 min window. Starting from trypsin-digested rNAPRT, a local spectral library was created to generate reference MS/MS spectra, including b- and y-fragment ions, and to determine the elution time for each peptide. Peptides were selected as unmodified proteotypic peptides with good signal stability and a wide range of elution times. Data processing was performed with Skyline software, freely available[45]. The identification of NAPRT was performed by extracting, post-acquisition, the chromatographic traces of specific fragment ions with acceptable purity, which have been subjected to an iterative selection. For each targeted peptide, extracted ion chromatograms (XIC) were selected for three to five transitions.

**RNA extraction and quantitative real-time PCR**. Quantitative real-time PCR (qRT-PCR) was performed as described[46]. TaqMan Gene Expression Assays (Thermo-Fisher) used: Hs01555410_m1 (*IL1B*), Hs01113624_g1 (*TNFA*), Hs00174103_m1 (*IL8*), Hs00234142_m1 (*CCL3*), Hs00354836 (*CASP1*), Hs00175721_m1 (*P2XR7*), Hs00174164_m1 (*CSF1*), Hs00152939_m1 (*TLR4*), Mm00441242_m1 (*Ccl2*), Mm00434228_m1 (*Il1b*), Mm00443258_m1 (*Tnf*). Hs00984230_m1 (*B2M*), and Mm02619580_g1 (*Actb*) were used as housekeeping genes.

**RNA sequencing**. Libraries were generated from total RNA of macrophages treated (6 h) with rNAPRT, rNAMPT (1 μg/ml) or LPS (2 μg/ml; Sigma-Aldrich) using the TruSeq RNA Sample Preparation v2 (Illumina, San Diego, CA). Samples were sequenced on the Illumina NextSeq 500 platform. Reads were mapped on the hg19 *Homo sapiens* reference assembly using TopHat v2.0.6 (Johns-Hopkins University, Baltimore, MD). Raw counts were computed using the featureCounts package[47] and the latest RefSeq annotation downloaded from the UCSC server (https://genome.ucsc.edu/cgi-bin/hgTables). Differential expression analysis was performed using R and the DESeq2 package[48]. Genes with abs (log₂(Fold Change)) ≥0.5, and $p < 0.05$ were retained for downstream analysis considered. Venn diagrams were created using Venny free on-line tools (http://bioinfogp.cnb.csic.es/tools/venny/) to visualize intersections between class comparison results and to select the sequences of interest. For enrichment analysis, differentially expressed genes were classified according to their gene ontology (GO) annotations using Database for Annotation, Visualization and Integrated Discovery (DAVID) Bioinformatics Resources (http://david.abcc.ncifcrf.gov/), and REVIGO (http://revigo.irb.hr/).

**Western blot and immunoprecipitation**. Cells lysates or immunoprecipitated fractions were resolved by SDS-PAGE and transferred to nitrocellulose membranes (Bio-Rad)[49]. Western blot reactions were visualized using ImageQuant LAS4000 and densitometric analyses performed using ImageQuantTL 7.0 software (GE Healthcare, Chicago, IL). Band intensity was quantified after normalizing over the corresponding unphosphorylated protein or over actin, used as loading control.

Anti-NAPRT monoclonal antibody (ProteinTech), chemically coated to luminex beads was used for immunoprecipitation of NAPRT from plasma samples. Immunoprecipitated fractions were resolved by SDS-PAGE and NAPRT detected

with a different antibody (MyBioSource). For detection of NAPRT in macrophage supernatants (SN), macrophages were plated for 24 h in RPMI + 0.1%FCS before western blot analysis.

**Surface plasmon resonance experiments**. SPR measurements, using MP-SPR Navi 210A VASA system (BioNavis), were used in order to confirm in vitro the TRL4/NAPRT interactions. For the measurements, 2D (planar) carboxymethyldextran (CMD) hydrogel-coated sensor slides (SPR102-CMD-2D, BioNavis) were chosen. The monoclonal antibodies against NAPRT purchased from Proteintech (www.ptglab.com) were immobilized on the carboxymethyldextran chip 2D (SPR102-CMD-2D) after activation with N-hydroxysuccinimide and N-ethyl-N-(3-diethylaminopropyl) carbodiimide (0.2 M EDC/0.05 M NHS). Anti-NAPRT (50 μg/ml) diluted in 10 mM sodium acetate buffer pH 5.0, was injected for immobilization on the CMD sensor surface. Non-reacted NHS ester groups were deactivated with 1 M ethanolamine pH 8.5 injection. The degassed running buffer for the immobilization process and TRL4/NAPRT measurements was of 10 mM HEPES, 0.05% Tween (pH 7.5). For binding experiment, the follow samples were fluxed on the functionalized surface: TRL4 (1 μM), NAPRT (100 nM) and the complex TRL4/NAPRT (1 μM/100 nM) diluted in running buffer. The injection time was 210 s at a flow rate of 30 μl/min. At the end of the binding the regeneration step occurred via addition of 10 mM glycine pH 2.0. SPR Navi control and SPR Navi Data Viewer/Trace Drower (BioNavis) were used to control the SPR measurements.

**Light microscopy**. Giemsa staining images were acquired using a CANON EOS 600D camera fitted to an AXIO Lab A1 ZEISS microscope.

**Confocal microscopy**. Cells on glass cover slips were rinsed, fixed, permeabilized, saturated, and stained with the indicated antibodies. Counter-staining was with AlexaFluor 568-comjugated phalloidin and DAPI. Fluorescence was acquired using a TCS SP5 laser scanning confocal microscope, using an oil immersion ×63 objective. Images were acquired with LAS AF software (both from Leica Microsystems, Milan, Italy). Files were processed with Photoshop (Adobe Systems, San Jose, CA). Pixel intensity was calculated using the ImageJ software (http://rsbweb.nih.gov/ij/).

**Cytokine/chemokine measurement**. IL8, TNFα, and CCL3 concentrations were determined using Bio-Plex/Luminex assays (Bio-Rad). IL-1β and M-CSF were determined using ELISA assays (Thermo-Fisher, Waltham, MA).

**Human and murine macrophage generation and treatment**. Peripheral blood mononuclear cells (PBMC) were seeded in 24-well plates ($10^7$ per well) in monocyte attachment medium (1 h, 37 °C PromoCell-GmbH, Heidelberg, Germany). Non-adherent cells were removed before adding RPMI + 10% FCS (Sigma-Aldrich, Saint Louis, MO) supplemented with recombinant human macrophage colony-stimulating factor (M-CSF; 50 ng/ml PeproTech, London, UK). Cell morphology and numbers were studied by Giemsa staining[50].

To exclude endotoxin contamination of rNAPRT preparations, macrophages were treated with: (i) boiled rNAPRT (100 °C, 10 min); (ii) digested rNAPRT (trypsin-EDTA 0.05%, 37 °C, overnight), (iii) inactivated LPS [polymyxin B, 100 μg/ml (Sigma-Aldrich, 4 °C, 1 h)].

For triggering of apoptosis macrophages were treated for 16 h with TNF-α (2 ng/ml; Peprotech) plus cycloheximide (35 μM; Sigma), while for necrosis 16 h with ionomycin (5 μM; Sigma) plus carbonyl cyanide m-chlorophenylhydrazone (CCCP, 20 μM; Sigma). Macrophage SN were collected and then 10× concentrated and loaded for WB.

For endotoxin tolerance experiments HD macrophages were preconditioned for 30 h with a sublethal dose of LPS (10 ng/ml) or rNAPRT (10 ng/ml), and then challenged with an acute dose of LPS (1 μg/ml) or rNAPRT (1 μg/ml) for 6 h. The negative control contained only medium, while the positive control received medium during the pre-conditioning phase but perceived an acute dose of LPS or rNAPRT during the challenging phase. Experiments were performed in RPMI 5% FCS.

TLR4$^{-/-}$ mice [B6(Cg)-Tlr4tm1.2Karp/J] or C57BL/6 wt were purchased from The Jackson Laboratory (Bar Harbor, ME). Macrophages were obtained from the peritoneal cavity and bone marrow, by the procedures described in refs. [51,52] and incubated for 6 days with 50 ng/ml of murine M-CFS (315-02, Peprotech, London, UK) and then treated as indicated.

**TLR4 and MyD88 silencing**. Differentiated macrophages were transfected with 200 nM of Silencer TLR4 Select Validated small interference RNA (siRNA, s14194) or 100 nM of MyD88 gene solution siRNA (#1027416, Qiagen, Venlo, NL) and Silencer Select Negative Control #1 siRNA (scramble, AM4611, Thermo-Fisher), using effectene transfection reagent (Qiagen). Cells were transfected in RPMI 10% FCS without changing the medium after transfection and analyzed after 72 h.

**Mice treatment**. Balb/c mice were bred in the animal facility at the Molecular Biotechnology Center, University of Turin. LPS from *E. coli* O55:B5 (25 mg/kg, Sigma) was injected intraperitoneally (i.p.) in male mice aged 6–8 weeks. rNAPRT

was injected i.p. at concentration of 1 or 25 mg/kg per mouse alone or 1–2 h before LPS treatment. Mice were closely monitored periodically to detect signs of endotoxemia and survival rate.

**Bioinformatics tools.** Molecular graphics and analyses were performed with the UCSF Chimera Package[53]. The 3D structures of HsNAPRT (PDB code: 4yub) was aligned to EfNAPRT (PDB code: 4mzy) and HsNAMPT (PDB code: 2gvg) using PROMALS3D[54] and the figures were produced using ESPript[55] EfNAPRT represents the closest homolog to SpNAPRT (73% sequence identity) with an available 3D structure and was thus used for structural comparison.

**Statistical analyses.** Continuous variables were compared by Mann–Whitney $U$ (unpaired data) or with unpaired $t$-test and Wilcoxon signed rank test or paired $t$-test (all two-tailed tests). Results are reported as box plots, where the top and bottom margins of the box define the 25th and 75th percentile, the line in the box defines the median and the error bars define the minimum and maximum of all data. In the scatter dot plots the line represents the median and the error bars define the interquartile range. $*p \leq 0.05$, $**p \leq 0.001$, $***p \leq 0.0001$, $****p \leq 0.0001$.

In order to identify the best NAPRT cut-off to maximize the marker prognostic power, we also developed an automatic R pipeline to deterministically test all the NAPRT thresholds. Thus, we computed a confusion matrix for each NAPRT threshold to perform significance tests and have data prepared in a convenient fashion for further risk ratio analysis. Each confusion matrix has been validated against Fisher exact test, to assess its statistical significance, and the pipeline produced an output file, which summarizes, for each cut-off, its risk ratio along with 95% confidence intervals and the $p$-value of the Fisher exact tests. We also trained a generalized linear model (GLM) to better understand combined effect among variables and to highlight possible NAPRT interactors, to further tune up its prognostic capabilities. The GLM model has been trained with a binomial link function. In order to explore the very large population produced after computing all the permutations among the effects, we took advantage of a genetic algorithm in order to increase efficiency and automate the selection of the top models. In order to obtain more reliable results, models have been ranked according to their Bayesian information criterion (BIC) score. BIC score is in fact more robust against overfitting, possibly induced when dealing with multiple combined effects[56].

**Reporting summary.** Further information on research design is available in the Nature Research Reporting Summary linked to this article.

## Data availability
All data generated or analyzed during this study are included in the Source Data file. For further request please contact the corresponding author. The RNA-seq data have been deposited in the Gene Expression Omnibus (GEO) database under the accession GSE135753.

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

## Acknowledgements

We wish to acknowledge clinicians who provided cancer patient/HD samples: Prof. G. Gaidano (University of Oriental Piedmont, Novara); Dr. M. Mandalà (Papa Giovanni XXIII Hospital, Bergamo); Dr. M. Coscia (University of Turin, Turin); Dr. F. Grosso and Dr. R. Libener (A.O.N. SS. Antonio e Biagio e Cesare Arrigo, Alessandria); Dr. L. Labanca, Dr. A. Bordiga, Dr. M. Airoldi, Dr. A. Palumbo and Dr. A. Zitella (A.O.U. Città della Salute e della Scienza, Turin); Dr. A. Naccarati and Dr. S. Tarallo (IIGM, Turin). We also wish to acknowledge Dr. N. Dani (Bioclarma, Turin) for excellent support in setting-up NAPRT bioplex assay and Prof. Federica Cavallo for providing Balb/c mice. This work was carried out in the laboratories of the Italian Institute for Genomic Medicine (IIGM). This work was supported by the Italian Institute for Genomic Medicine Institutional funds, by the Universities of Torino and Polytechnic University of Marche research grants, by the Associazione Italiana per la Ricerca sul Cancro AIRC, by the Gilead Fellowship Program 2018 and by Ministero Istruzione e Università, Progetto Eccellenza 2018-2022 and PRIN project 2017CBNCYT.

## Author contributions

A.M. and V.A. designed and performed experiments, interpreted results, and contributed to writing the paper; F.M. and G.M. prepared recombinant proteins and measured enzyme activities; L.S. performed protein structural and bioinformatics analysis; A.V. and S.D.A. performed SPR analyses; F.G., K.G., and G.M. performed biochemical assays; N.V. helped in performing in vivo experiments; D.I. performed RNA-seq experiments; A.I. and G.M. provided septic patient samples; G.P. performed statistical analysis; G.M. and S.O. interpreted data; N.R. and S.D. designed the study, interpreted data, and wrote the paper

## Additional information

**Competing interests:** The authors declare no competing interests.

