## [Peer Review File · Nature Communications]

Reviewers' comments:

Reviewer #1 (TLR signalling, ubiquitin)(Remarks to the Author):

In this paper the authors report that the enzyme nicotinate phosphoribosyltransferase (NAPRT), which is present in the blood, can activate Toll-Like Receptor 4 (TLR4) in human macrophages and trigger the activation of NF- κ B and the production of mRNAs encoding inflammatory mediators. The rationale for conducting these experiments appears to be that another enzyme that participates in NAD synthesis, nicotinamide phosphoribosyltransferase, is also present in the blood and has been reported to act in a similar way. However, the effect of NAPRT occurs independently of its enzymatic activity. I have the following comments this paper: -

Major points

1. The recombinant NAPRT (rNAPRT) used to stimulate macrophages, was produced in bacteria. I do not see any data in the paper, which excludes the possibility that the effects of rNAPRT are due to contamination with endotoxin. There are standard ways of excluding this possibility which need to be carried out
2. The concentration of rNAPRT used to activate macrophages in nearly all the experiments was 1.0 μ g/ml, which is about 500 times the concentration of NAPRT found in the blood of human donors, so even if the effect of rNAPRT is not caused by contamination with endotoxin, the physiological significance seems obscure. In patients with sepsis, the concentration of NAPRT in the serum was higher, but this is not surprising in view of the extensive cell death and release of their contents under these conditions. Incidentally the concentration of LPS used by the authors in control studies to stimulate macrophages (2 μ g/ml) is huge and 100 times higher than the concentration needed to elicit a near maximal effect.

Other comments

3. This reviewer did not find the experiments in which TLR4 was reduced by siRNA to be at all convincing. The knock-down was only partial and the effects were not shown to be reversed by transfection with an siRNA-resistant mutant TLR4. It would be much more convincing to knock-out TLR4 completely in immortalized human macrophages using CRISPR/Cas9 gene editing technology, or to conduct experiments in primary macrophages from TLR4 KO mice followed by stimulation with human and/or mouse rNAPRT
4. The authors used the phosphorylation of p65 at Ser536 as their readout of NF- κ B activation. More commonly used readouts, such as the degradation of I κ B α and/or the phosphorylation of IKK β would be more convincing. It is also important to investigate whether the phosphorylation of p65 was prevented by specific inhibitors of IKK β .

Reviewer #2 (NF κ b, TLR signalling)(Remarks to the Author):

The manuscript by Manago et al proposes extracellular nicotinate phosphoribosyltransferase (NAPRT) as a novel modulator of inflammatory responses by triggering the TLR4 pathway. Whilst the study presents some interesting initial findings, it is at a very preliminary stage and would require much more validation and comprehensive analysis. Outstanding questions are many and include:

1. Are the preparations of rNAPRT contaminated with endotoxin/LPS, a frequent problem with recombinant protein generation. Indeed much of the TLR4-dependent bioactivity may be accounted for by such contamination.
2. Is NAPRT a novel DAMP and if so where is it coming from? (eg. from which cells and how is it released? eg is it released following various forms of cells death?

3. However does rNAPRT alone promote production of mature IL-1 β ? Does it activate inflammasomes
4. Does NAPRT show any in vivo inflammatory effects when administered to mice? Are these effects lost in TLR4 knockout/deficient mice.
5. Does NAPRT affect Myd88 dependent/independent signalling? (much more detailed analysis required of these pathways.)
6. Does NAPRT show tolerance with LPS.
7. Some specific issues relate to data eg. Figure 1E: It is not clear why IP of recombinant NAPRT-GST migrates with same mobility as the IP endogenous NAPRT questioning the identity of these immunoreactive bands. Furthermore, the inclusion of control IP experiments using non-immune IgG-beads would be valuable.

Point-by-point reply to reviewers

Reviewer #1:

In this paper the authors report that the enzyme nicotinate phosphoribosyltransferase (NAPRT), which is present in the blood, can activate Toll-Like Receptor 4 (TLR4) in human macrophages and trigger the activation of NF- κ B and the production of mRNAs encoding inflammatory mediators. The rationale for conducting these experiments appears to be that another enzyme that participates in NAD synthesis, nicotinamide phosphoribosyltransferase, is also present in the blood and has been reported to act in a similar way. However, the effect of NAPRT occurs independently of its enzymatic activity. I have the following comments this paper:

Major points

1. The recombinant NAPRT (rNAPRT) used to stimulate macrophages, was produced in bacteria. I do not see any data in the paper, which excludes the possibility that the effects of rNAPRT are due to contamination with endotoxin. There are standard ways of excluding this possibility which need to be carried out.

Author's reply: Thanks for pointing out this issue. Following reviewer's recommendation, we performed new experiments aimed at excluding the possibility of LPS contamination in our preparations. Specifically:

- i) We boiled rNAPRT confirming complete loss of NF- κ B activation, at variance with what observed in the case of LPS.*
- ii) To confirm that loss of protein structure results in loss of NF- κ B activation, we digested rNAPRT using trypsin, confirming loss of NF- κ B activation.*

These results are now presented in Figure 2I and discussed in the text.

There are two additional reasons why we believe the results obtained using rNAPRT are not due to LPS contamination. The first one is that we used Clear Coli to produce recombinant proteins and the second is that we have mutants, both of NAMPT and NAPRT, that are produced in the same way, but fail to trigger NF- κ B activation. Both wild-type and mutant NAMPT and NAPRT recombinant proteins were used under the same experimental conditions to treat macrophages and p65 phosphorylation used as readout. In addition, bacterial NAPRT and NAMPT, also produced in the same way fail to activate signaling. Our argument is that if the effects were simply due to LPS contamination, that would likely be seen in all recombinant protein preparations.

2. The concentration of rNAPRT used to activate macrophages in nearly all the experiments was 1.0 μ g/ml, which is about 500 times the concentration of NAPRT found in the blood of human donors, so even if the effect of rNAPRT is not caused by contamination with endotoxin, the physiological significance seems obscure. In patients with sepsis, the concentration of NAPRT in the serum was higher, but this is not surprising in view of the extensive cell death and release of their contents under these conditions. Incidentally the concentration of LPS used by the authors in control studies to stimulate macrophages (2 μ g/ml) is huge and 100 times higher than the concentration needed to elicit a near maximal effect.

Author's reply: The concentration of 1 microgram/ml for NAPRT was selected as it was the concentration used for NAMPT (Camp S.M. et al. 2015 PMID: 26272519) and for the majority of endogenous TLR4 ligands in similar experiments (Kim S. et al 2013 PMID:23508573; Fan J. et al. 2007 PMID:17475888; Mendez O. et al 2018 PMID:30135148). The second consideration is that we have measured plasmatic concentrations of NAPRT, which may be very different from local concentrations. In any case, following the reviewer's suggestion, we have performed dose titration experiments using macrophages from 5 new donors, starting at 1 microgram/ml to 31 ng/ml of

NAPRT. The results clearly show a dose-dependent effect, with p-p65 still being visible at 31 ng/ml, while pIKK α /b seems to require higher levels of rNAPRT. These results are now included as Figure 2G.

We are aware that LPS was dosed at extremely high levels. We used it as internal control and wanted to be in a condition to observe maximal activation of TLR4 following a short treatment. For this reason, after searching the literature, we settled for 1-2 micrograms/ml.

In long term differentiation experiments, the dose of LPS, NAPRT and NAMPT was reduced to 10 nanograms/ml, in order to be in a more physiological situation.

Other comments

3. This reviewer did not find the experiments in which TLR4 was reduced by siRNA to be at all convincing. The knock-down was only partial and the effects were not shown to be reversed by transfection with an siRNA-resistant mutant TLR4. It would be much more convincing to knock-out TLR4 completely in immortalized human macrophages using CRISPR/Cas9 gene editing technology, or to conduct experiments in primary macrophages from TLR4 KO mice followed by stimulation with human and/or mouse rNAPRT.

Author's reply: We agree with the reviewer that the silencing experiment was only partly convincing. For this reason, we followed the approach suggested by the reviewer, i.e. we used macrophages obtained from TLR4 KO mice. In order to do this, we first showed that human NAPRT is able to activate mouse macrophages and then repeated the experiment using macrophages from TLR4 KO animals. Experimental readouts included short-term results, such as activation of NF- κ B, and long-term ones, such as cytokine secretion. In both cases, no effects were observed when macrophages from TLR4 KO mice were challenged with rNAPRT. Signaling in response to LPS represented the control experiment. Results are now reported in Figure 4G-H.

Furthermore, to obtain evidence of a direct interaction between TLR4 and NAPRT we used surface plasmon resonance. For this experiment, we reproduced the experimental conditions used to show interaction between NAMPT and TLR4. Specifically, an anti-NAPRT antibody was coated to the surface before incubation with rNAPRT or with a mixture of rNAPRT and TLR4. Results indicate that the mixture of NAPRT and TLR4 shows increased binding, indicating a direct molecular interaction. Results are now shown in Figure 4A.

Lastly, we wanted to identify the region(s) of NAPRT responsible for the interaction with TLR4. In silico predictions highlighted a mouth-like regions, framed by positively charged residues, as a potential candidate. Substitution of the positive amino acids with neutral ones, completely inhibited TLR4 activation, while maintaining a correctly folded protein. These results are now shown in Figure 5F-G.

4. The authors used the phosphorylation of p65 at Ser536 as their readout of NF- κ B activation. More commonly used readouts, such as the degradation of I κ B α and/or the phosphorylation of IKK β would be more convincing. It is also important to investigate whether the phosphorylation of p65 was prevented by specific inhibitors of IKK β .

Author's reply: Phosphorylation of IKK α and ERK1/2 were added as a functional readout to NAPRT challenge. Results are reported in Figure 2D.

Reviewer #2 (Remarks to the Author):

The manuscript by Manago et al proposes extracellular nicotinate phosphoribosyltransferase (NAPRT) as a novel modulator of inflammatory responses by triggering the TLR4 pathway. Whilst the study presents some interesting initial findings, it is at a very preliminary stage and would

require much more validation and comprehensive analysis. Outstanding questions are many and include:

1. Are the preparations of rNAPRT contaminated with endotoxin/LPS, a frequent problem with recombinant protein generation. Indeed much of the TLR4-dependent bioactivity may be accounted for by such contamination.

Author's reply: There are two main reasons why we believe the results obtained using rNAPRT are not due to LPS contamination. The first one is that we used Clear Coli to produce recombinant proteins and the second is that we have mutants, both of NAMPT and NAPRT, that are produced in the same way, but fail to trigger NF- κ B activation. However, following the reviewer's recommendation, we performed new experiments aimed at excluding the possibility of LPS contamination in our preparations. Specifically:

- i) We boiled rNAPRT confirming complete loss of NF- κ B activation, at variance with what observed in the case of LPS.*
- ii) To confirm that loss of protein structure results in loss of NF- κ B activation, we digested rNAPRT using trypsin, obtaining the same result as from boiling it.*

These results are now presented in Figure 2I and discussed in the text.

2. Is NAPRT a novel DAMP and if so where is it coming from? (eg. from which cells and how is it released? eg is it released following various forms of cells death

Author's reply: At present we only know that NAPRT can be dosed in the supernatants of viable human macrophages. We do not know how the molecule reaches the extracellular environment or whether there are other cells that can release NAPRT. This is a common situation for many DAMPS, including NAMPT (or visfatin), the most studied NAD-biosynthetic enzyme. In general, the working hypothesis is that DAMPS are not simply released from dead or dying cells, but that there are active mechanisms behind their presence in the extracellular environments. The molecular nature behind these mechanisms is currently being tested.

3. However does rNAPRT alone promote production of mature IL-1 β ? Does it activate inflammasomes

Author's reply: RNAseq data showed that rNAPRT (1 μ g/ml) treatment for 6 hours of human macrophages induces transcriptional up-regulation of several cytokines linked to activation of the inflammasome, including IL1B, IL8, TNFA. We confirmed these results also by RT-PCR (Supplementary Figure 3A-C). All these cytokines were dosed as secreted proteins using ad hoc ELISA/luminex assays. The ELISA for IL1B detect the mature human protein and was purchased from Thermo Scientific. In favor of a role for NAPRT in activating the inflammasome, RNA-seq data showed CASP1 and P2XR7 induction (see data below, rpkm levels). Confirmation was obtained RT-PCR performed on cells from at least 6 different donors and clearly showing up-regulation of CASP1 and P2XR7 mRNA following treatment with NAPRT (Supplementary Figure 3C).

Symbol	NT	NAPRT
IL1B	117,8892	2519,79733
IL8	245,0698	2385,002
TNF	48,72733	125,974667
P2RX7	41,22295	84,4727
CASP1	26,15515	108,324

4. Does NAPRT show any in vivo inflammatory effects when administered to mice? Are these effects lost in TLR4 knockout/deficient mice.

Author's reply: we used macrophages obtained from TLR4 KO mice. In order to do this, we first showed that human NAPRT is able to activate mouse macrophages and then repeated the experiment using macrophages from TLR4 KO animals. Experimental readouts included short-term results, such as activation of NF- κ B, and long-term ones, such as cytokine secretion. In both cases, no effects were observed when macrophages from TLR4 KO mice were challenged with rNAPRT. Signaling in response to LPS represented the control experiment. Results are now reported in Figure 4G-H.

These data confirm that TLR4 is needed for NAPRT to signal. In order to obtain evidence of a direct interaction between TLR4 and NAPRT we used surface plasmon resonance. For this experiment, we reproduced the experimental conditions used to show interaction between NAMPT and TLR4. Specifically, an anti-NAPRT antibody was coated to the surface before incubation with rNAPRT or with a mixture of rNAPRT and TLR4. Results indicate that the mixture of NAPRT and TLR4 shows increased binding, indicating a direct molecular interaction. Results are now shown in Figure 4A. Lastly, we wanted to identify the region(s) of NAPRT responsible for the interaction with TLR4. In silico predictions highlighted a mouth-like regions, framed by positively charged residues, as a potential candidate. Substitution of the positive amino acids with neutral ones, completely inhibited TLR4 activation, while maintaining a correctly folded protein. These results are now shown in Figure 5F-G.

5. Does NAPRT affect Myd88 dependent/independent signalling? (much more detailed analysis required of these pathways.)

Author's reply: RNAseq data showed that in macrophages the activation of TLR4 signaling via NAPRT could be MYD88-dependent. There is an up-regulation of MYD88, IRAK1, IRAK4 levels, while molecules involved in the MYD88-independent pathway, including TRIF, TBK1 and IRF3 are expressed at very low levels and not significantly modulated in response to NAPRT exposure. RNA-seq data (rpkm values) are included below for your evaluation.

Symbol	NT	NAPRT
MYD88	75,1818	118,284
IRAK1	38,2405	71,7253
IRF3	8,49047	6,50316
TICAM1(TRIF)	10,4713	14,1751
TBK1	18,6057	15,7267

6. Does NAPRT show tolerance with LPS.

Author's reply: Following the observations of the reviewer, we attempted to answer this question, using in vitro models. To do so, human macrophages were challenged with low dose NAPRT, before using a maximal dose of LPS. The synthesis of TNF-alpha was used as experimental readout. Results are now reported in Figure 6A and show that pre-treatment with NAPRT decreases responses to a maximal dose of LPS, suggesting that NAPRT mediates endotoxin tolerance.

7. Some specific issues relate to data eg. Figure 1E: It is not clear why IP of recombinant NAPRT-GST migrates with same mobility as the IP endogenous NAPRT questioning the identity of these immunoreactive bands. Furthermore, the inclusion of control IP experiments using non-immune IgG-beads would be valuable.

Author's reply: To confirm that the protein detected by luminex is NAPRT, two plasma samples containing high levels of the enzyme were immunoprecipitated using an anti-NAPRT monoclonal antibody covalently bound to immunomagnetic beads used in the luminex assay, revealing a band with a molecular weight of \approx 58 kDa, compatible with the NAPRT monomer.

We loaded in the gel as controls: i) rNAPRT-His-tag (home-made recombinant full-length protein), first lane. The His-tag is in the range of 1-2 kDa; ii) IP of rNAPRT-GST-tag (MyBioSource MBS969577). This recombinant protein was used to build the titration curve of the luminex assay and it comprises only the final 310 amino acids (see diagram below). The predicted molecular weight is 33199 Da. GST-tag is a 211 amino acid protein (26 kDa). The sum of NAPRT partial protein and GST-tag is around 59 kDa, thus very similar to endogenous NAPRT and undistinguishable under our experimental conditions.

hNAPRT SEQUENCE 538 AA; 57578 MW;

```

MAAEQDPEAR AAARPLLTDL YQATMALGYW RAGRARDAAE FELFFRRCPF GGAFALAAGL
RDCVRFLRAF RLRDADVQFL ASVLPPDTPD AFFEHLRALD CSEVTVRALP EGSLAFPGVP
LLQVSGPLLV VQLLETPLLC LVSYASLVAT NAARLRLIAG PEKRLLEMGL RRAQGPDGGL
TASTYSYLG GFDSSSNVLAG QLRGVPVAGT LAHSFVTSFS GSEVPPDPML APAAGEGPGV
DLAAKAQVWL EQVCAHLGLG VQEPHPGERA AFVAYALAFP RAFQGLLDY SVWRSGLPNF
LAVALALGEL GYRAVGVRD SGDLLQQAQE IRKVFRAAAA QFQVPWLESV LIVVSNNDI
EALARLAQEG SEVNVIGIGT SVVTCPPQPS LGGVYKLVAV GGQPRMCLTE DPEKQTLPGS
KAAFRLGSD GSPLMDMLQL AEEPVPQAGQ ELRVWPPGAQ EPCTVRPAQV EPLLRCLCQQ
GQLCEPLPSL AESRALAQLS LSRLSPEHRR LRSPAQYQVV LSERLQALVN SLCAGQSP

```

NAPRT MyBiosource portion

iii) The primary anti-NAPRT antibody used for immunoprecipitation, loaded in the last right lane and recognized by secondary antibody, was an internal control to confirm that samples were not contaminated by primary antibody.

For a different set of IP experiments using different magnetic beads we used an irrelevant antibody (mouse-IgG_{total}) demonstrating no visible band in the lane (IP IRR). Enclosed image also for your evaluation.

IB: NAPRT NOVUS (rabbit)

Reviewers' comments:

Reviewer #2 (Remarks to the Author):

The revised manuscript by Manago et al addresses some of the technical issues raised in the primary review. However, it is very disappointing that the authors did not address the substantive and important concerns highlighted by this reviewer. The following serious concerns remain:

1. Is NAPRT a novel DAMP and if so where is it coming from? No attempts have been to address this question, its cell sources, conditions under which it is released (eg. different forms of cells death?). These data are extremely important to provide meaningful support for NAPRT being a novel DAMP. The authors indicate that these questions are being tested but data from these experiments are required for this manuscript.
2. Whilst the revised manuscript reports effects of rNAPRT on expression levels of inflammasome components, there are no efforts to directly study activation of inflammasome pathways and the effects of rNAPRT.
3. No efforts have been made to explore if NAPRT shows any in vivo inflammatory effects when administered to mice and if these effects are lost in TLR4 knockout/deficient mice. Some data are included on ex vivo cells from these mice but the authors should at the very least report the effects of NAPRT when administered to these mice.
4. Little effort has been made to directly address if NAPRT affects Myd88 dependent/independent signalling? Analysis of the expression levels of components of these pathways is not a direct measure of their activation.

We would like to thank Reviewer #3 for her/his positive evaluation of our work.

Dear Reviewer #2,

Thanks for the evaluation of our work. Following is a point-by-point reply to the issues raised.

Reviewer's issue: *Is NAPRT a novel DAMP and if so where is it coming from? No attempts have been to address this question, its cell sources, conditions under which it is released (eg. different forms of cells death?). These data are extremely important to provide meaningful support for NAPRT being a novel DAMP. The authors indicate that these questions are being tested but data from these experiments are required for this manuscript.*

Authors' reply: In the revised version, we have followed your suggestion and have examined whether different forms of cell death modulate NAPRT levels in the extracellular space. As all our work is based on the analysis of normal human macrophages, we have induced apoptosis and necrosis of these cells, by using approaches published by the Bianchi group when studying HMGB1. Results reported in the revised Figure 6 show that necrosis induces massive release of NAPRT, indirectly substantiating the observation that in patients with septic shock the plasmatic levels of the enzyme are markedly elevated (data shown in revised Fig S8).

Reviewer's issue: *Whilst the revised manuscript reports effects of rNAPRT on expression levels of inflammasome components, there are no efforts to directly study activation of inflammasome pathways and the effects of rNAPRT.*

Authors' reply: In the revised version we show activation of the inflammasome by western blot analysis of caspase 1 and NLPR3, (revised Figure 2, with blots quantified in the revised Supplementary Figure 3).

Reviewer's issue: *No efforts have been made to explore if NAPRT shows any in vivo inflammatory effects when administered to mice and if these effects are lost in TLR4 knockout/deficient mice. Some data are included on ex vivo cells from these mice but the authors should at the very least report the effects of NAPRT when administered to these mice.*

Authors' reply: We thank the reviewer for raising this issue, which is very interesting and represents the starting point of our follow-up work. To address this issue, we started from our observation that NAPRT lowered LPS-induced cytokine responses *in vitro* in normal human macrophages. Following this observation, we asked what happens *in vivo* if we administer NAPRT with different doses of LPS. Firstly, NAPRT administered alone at "high" doses (25 mg/kg) induces a phenotype similar to what observed with LPS, even if significantly less severe. No mouse died following NAPRT administration. No effects could be observed when treating at "low" doses (1 mg/kg). However, the interesting finding is that low doses of NAPRT prevent death when LPS is administered at lethal concentrations (25 mg/kg), recapitulating our *in vitro* observations of endotoxin tolerance. Even if this is a whole new project in itself, we have inserted this preliminary observation in the revised Figure 6.

Reviewer's issue: *Little effort has been made to direct address if NAPRT affects Myd88 dependent/independent signaling? Analysis of the expression levels of components of these pathways is not a direct measure of their activation.*

Authors' reply: To address this issue we have used a dual approach. On the one hand, we have looked for activation of signaling intermediates that are specific for the MyD88 pathway, while on the other we have silenced MyD88 in normal human macrophages and studied NAPRT signaling in these conditions.

In the revised Figure 2 we show degradation of IRAK1 following NAPRT exposure in normal human macrophages, an accepted indications that the signaling proceeds via MyD88. Furthermore, silencing MyD88 in normal human macrophages was followed by significant loss of NF- κ B activation following incubation with NAPRT, further substantiating the role of MyD88 in this setting.

REVIEWERS' COMMENTS:

Reviewer #2 (Remarks to the Author):

The authors have shown meaningful and worthwhile efforts in adequately addressing all of the issues raised in the last phase of review.